# Guidance in the Frequency Domain Enables High-Fidelity Sampling at Low CFG Scales

## Abstract

Classifier-free guidance (CFG) has become an essential component of modern conditional diffusion models. Although highly effective in practice, the underlying mechanisms by which CFG enhances quality, detail, and prompt alignment are not fully understood. This paper presents a novel perspective on CFG by analyzing its effects in the frequency domain, showing that low and high frequencies have distinct impacts on generation quality. Specifically, low-frequency guidance governs global structure and condition alignment, while high-frequency guidance mainly enhances visual fidelity. However, applying a uniform scale across all frequencies—as is done in standard CFG—leads to oversaturation and reduced diversity at high scales and degraded visual quality at low scales. Based on these insights, we propose frequency-decoupled guidance (FDG), an effective approach that decomposes CFG into low- and high-frequency components and applies separate guidance strengths to each component. FDG improves image quality at low guidance scales and avoids the drawbacks of high CFG scales by design, i.e., retaining the benefits of CFG over unguided generation without its common failures. Through extensive experiments across multiple datasets and models, we demonstrate that FDG consistently enhances sample fidelity while preserving diversity, leading to improved FID and recall compared to CFG, establishing our method as a plug-and-play alternative to standard classifier-free guidance.

CFG                                   FDG (Ours)

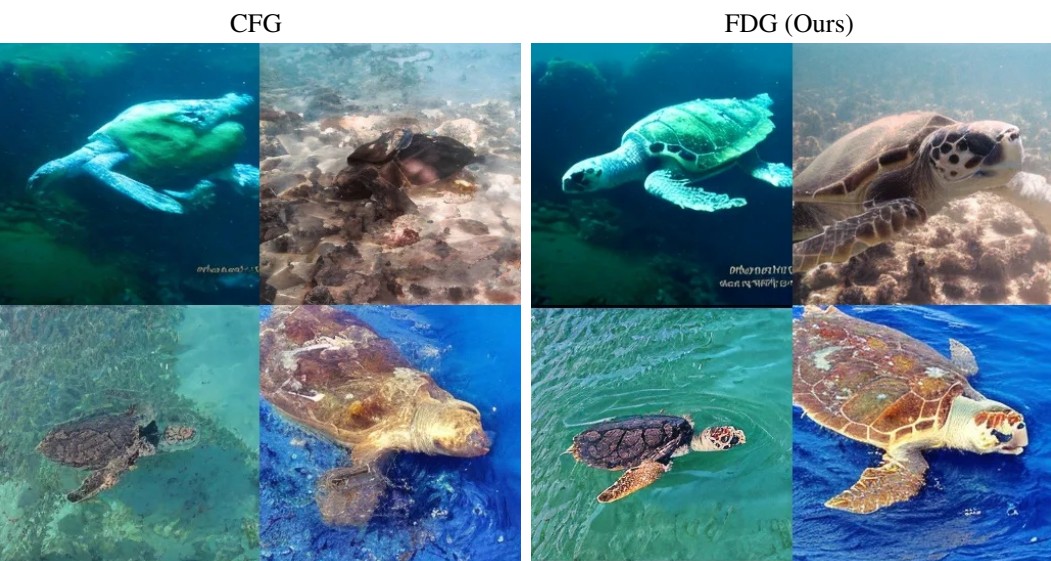

Figure 1: Low classifier-free guidance produces images with good diversity and color composition but often results in low-quality and blurry generations. We propose frequency-decoupled guidance (FDG), a novel modification to the CFG update rule in the frequency domain that significantly improves image quality at low guidance scales while, by design, avoiding common issues of high CFG scales such as reduced diversity. The examples here are generated using a class-conditional DiT-XL/2 model (Peebles & Xie, 2022) with a CFG scale of 1.2.

# 1 INTRODUCTION

Diffusion models (Sohl-Dickstein et al., 2015; Ho et al., 2020; Song et al., 2021b) are a class of generative models that learn the data distribution by reversing a forward noising process that progressively corrupts data with increasing levels of Gaussian noise. While the theory suggests that simulating this reverse process should generate high-quality samples, in practice, unguided sampling often produces low-quality images that poorly match the input condition. To mitigate this, classifier-free guidance (CFG) (Ho & Salimans, 2022) has become a standard technique in modern diffusion models for improving quality and prompt alignment—though often at the cost of reduced diversity (Ho & Salimans, 2022; Sadat et al., 2024a) and excessive oversaturation (Sadat et al., 2025).

Current diffusion models typically rely on high guidance scales to achieve better image quality and prompt alignment. However, high guidance scales degrade sample diversity and introduce color saturation artifacts (Ho & Salimans, 2022). Conversely, low CFG scales tend to produce more diverse samples with natural color compositions but often suffer from poor global structure and lower visual fidelity. To address these trade-offs, several empirical strategies have been proposed to balance diversity and quality of CFG (Kynkäänniemi et al., 2024; Sadat et al., 2024a; 2025; WANG et al., 2024). Despite this progress, a systematic understanding of how CFG improves image quality and prompt alignment remains limited. Existing works neither fully explain the internal mechanisms of CFG nor explore how to improve generation quality at low guidance scales.

In this paper, our objective is to advance the understanding of how CFG works and improve image quality at low CFG scales, thereby avoiding the detrimental effects associated with high guidance scales. We begin by analyzing the CFG update rule in the frequency domain and show that CFG enhances image quality and prompt alignment through distinct frequency components. Specifically, we find that the low-frequency components of the CFG signal primarily govern global structure and condition alignment, while its high-frequency components mainly contribute to visual quality and details, with minimal impact on the overall composition of the image. We also observe that excessive guidance in the low-frequency domain leads to reduced diversity and oversaturation, whereas high-frequency components usually benefit from higher guidance scales. Since standard CFG applies a uniform scale across all frequencies, this explains why low CFG scales degrade quality, and high CFG scales boost detail at the cost of diversity and oversaturation.

Building on this insight, we propose a CFG scheme in the frequency domain, termed frequency-decoupled guidance (FDG), which disentangles the guidance scales applied to the low- and high-frequency components of the CFG update. We argue that low-frequency components should be guided more conservatively to avoid low-diversity and oversaturated generations, while high-frequency components can benefit from stronger guidance to enhance image quality. By assigning separate guidance strengths to these components, FDG improves image quality while retaining the diversity typically associated with low CFG scales. FDG is among the first approaches to systematically enhance the quality of low CFG scales, thereby avoiding the drawbacks of high CFG scales by design.

FDG introduces practically no additional sampling cost and can be applied to any pretrained diffusion model without extra training or fine-tuning. Through extensive experiments, we show that FDG consistently improves image quality and maintains diversity across a range of datasets, models, and metrics. In other words, FDG delivers the quality benefits of CFG over unguided generation while avoiding its usual drawbacks, such as reduced diversity. Moreover, FDG offers a deeper understanding of how CFG enhances image quality and prompt alignment by isolating the roles of low- and high-frequency components in the CFG update throughout the inference process. As such, we consider FDG as a superior plug-and-play alternative to the standard classifier-free guidance.

# 2 RELATED WORK

Score-based diffusion models (Song & Ermon, 2019; Song et al., 2021b; Sohl-Dickstein et al., 2015; Ho et al., 2020) learn data distributions by reversing a forward process that progressively corrupts data with Gaussian noise. They have rapidly surpassed prior generative techniques in fidelity and diversity (Nichol & Dhariwal, 2021; Dhariwal & Nichol, 2021), achieving state-of-the-art results across unconditional image synthesis (Dhariwal & Nichol, 2021; Karras et al., 2022; Yu et al., 2025a; Karras et al., 2024), text-to-image generation (Podell et al., 2023; Esser et al., 2024; Qin et al., 2025), video synthesis (Blattmann et al., 2023b;a; Bar-Tal et al., 2024; Wan et al., 2025), image-to-image

translation (Saharia et al., 2022a; Liu et al., 2023a; Xia et al., 2023), and audio synthesis (Chen et al., 2021; Huang et al., 2023; Liu et al., 2023b; Tian et al., 2025).

Recent studies have introduced numerous enhancements to the original DDPM framework (Ho et al., 2020), including improved network architectures (Hoogeboom et al., 2023; Karras et al., 2023; Peebles & Xie, 2022; Dhariwal & Nichol, 2021), novel sampling strategies (Song et al., 2021a; Karras et al., 2022; Liu et al., 2022b; Lu et al., 2022a; Salimans & Ho, 2022), and better training techniques (Nichol & Dhariwal, 2021; Karras et al., 2022; Song et al., 2021b; Salimans & Ho, 2022; Rombach et al., 2022). Despite these advancements, guidance methods—such as classifier-free guidance (Ho & Salimans, 2022)—remain essential for enhancing sample quality and improving alignment between conditioning information and generated outputs (Nichol et al., 2022).

Although several recent works have focused on improving diffusion model training for high-quality unguided generation (Karras et al., 2023; Yu et al., 2025b; Leng et al., 2025; Hoogeboom et al., 2024), modern diffusion models still rely on high guidance scales to enhance image quality and prompt alignment. However, this comes at the cost of reduced diversity (Ho & Salimans, 2022; Sadat et al., 2024a) and undesirable artifacts such as oversaturation (Sadat et al., 2025). Our work addresses this issue by disentangling the benefits of high CFG from its drawbacks. Leveraging frequency decomposition, we introduce a novel approach that improves image quality at lower CFG scales to avoid the trade-offs associated with high CFG scales. In other words, FDG enhances quality relative to unguided generation as the guidance scale increases, while avoiding the diversity loss and oversaturation effects of CFG.

Frequency decomposition techniques, such as Laplacian pyramids (Burt & Adelson, 1983) and wavelet transforms (Brewster, 1993), have recently been leveraged in generative models to enhance both quality and efficiency (Denton et al., 2015; Gal et al., 2021; Atzmon et al., 2024; Sadat et al., 2024b; Agarwal et al., 2025; Xiao et al., 2024). However, their potential for improving the sampling behavior of diffusion models remains underexplored. We demonstrate that applying guidance in the frequency domain unifies the advantages of both high and low CFG regimes (i.e., the fidelity of high scales and the diversity and color balance of low scales) into a single, inference-only method.

## 3 BACKGROUND

**Diffusion models** Let $\boldsymbol{x} \sim p_{\text{data}}(\boldsymbol{x})$ denote a data sample, and let $t \in [0, 1]$ represent a continuous time variable. The forward diffusion process adds noise to the data as $\boldsymbol{z}_t = \boldsymbol{x} + \sigma(t)\boldsymbol{\epsilon}$, where $\sigma(t)$ is a time-dependent noise schedule. This schedule controls the degree of corruption, with $\sigma(0) = 0$ (no noise) and $\sigma(1) = \sigma_{\text{max}}$ (maximum noise). As shown by Karras et al. (2022), this forward process corresponds to the following ODE:

$$\mathrm{d}\boldsymbol{z} = -\dot{\sigma}(t)\sigma(t)\,\nabla_{\boldsymbol{z}_t} \log p_t(\boldsymbol{z}_t)\mathrm{d}t, \tag{1}$$

where $p_t(\boldsymbol{z}_t)$ is the distribution over noisy samples at time $t$, with $p_0 = p_{\text{data}}$ and $p_1 = \mathcal{N}\left(\boldsymbol{0}, \sigma_{\text{max}}^2 \boldsymbol{I}\right)$. Given access to the time-dependent score function $\nabla_{\boldsymbol{z}_t} \log p_t(\boldsymbol{z}_t)$, one can sample from the original data distribution by integrating the ODE in reverse from $t = 1$ to $t = 0$. Since this score function is unknown, it is approximated by a neural denoiser $D_{\boldsymbol{\theta}}(\boldsymbol{z}_t, t)$, which is trained to recover the clean data $\boldsymbol{x}$ from its noisy counterpart $\boldsymbol{z}_t$. Conditional generation is enabled by augmenting the denoiser with an auxiliary input $\boldsymbol{y}$, such as class labels or text prompts, yielding $D_{\boldsymbol{\theta}}(\boldsymbol{z}_t, t, \boldsymbol{y})$.

**Classifier-free guidance** Classifier-free guidance (CFG) is an inference technique designed to enhance the quality of generated samples by interpolating between conditional and unconditional model predictions (Ho & Salimans, 2022). Let $\boldsymbol{y}_{\text{null}} = \varnothing$ denote a null condition representing the unconditional case. CFG modifies the denoiser output at each sampling step as follows:

$$\hat{D}_{\text{CFG}}(\boldsymbol{z}_t, t, \boldsymbol{y}) = D_{\boldsymbol{\theta}}(\boldsymbol{z}_t, t, \boldsymbol{y}_{\text{null}}) + w(D_{\boldsymbol{\theta}}(\boldsymbol{z}_t, t, \boldsymbol{y}) - D_{\boldsymbol{\theta}}(\boldsymbol{z}_t, t, \boldsymbol{y}_{\text{null}})), \tag{2}$$

Here, $w = 1$ corresponds to the unguided case (i.e., sampling with $D_{\boldsymbol{\theta}}(\boldsymbol{z}_t, t, \boldsymbol{y})$). The unconditional model $D_{\boldsymbol{\theta}}(\boldsymbol{z}_t, t, \boldsymbol{y}_{\text{null}})$ is typically trained by randomly replacing the condition $\boldsymbol{y}$ with $\boldsymbol{y}_{\text{null}}$ during training. Similar to the truncation trick in GANs (Brock et al., 2019), CFG improves image quality, but often at the cost of sample diversity (Murphy, 2023) and oversaturation (Sadat et al., 2025).

**Frequency decompositions** Multi-level frequency decompositions, such as Laplacian pyramids and wavelet transforms, are commonly used to separate an image into different frequency bands.

$w_{\text{low}} = 1.5, w_{\text{high}} = 1.5$  $w_{\text{low}} = 7, w_{\text{high}} = 7$  $w_{\text{low}} = 7, w_{\text{high}} = 1.5$  $w_{\text{low}} = 1.5, w_{\text{high}} = 7$

low quality, high diversity  high quality, low diversity  low quality, low diversity  high quality, high diversity

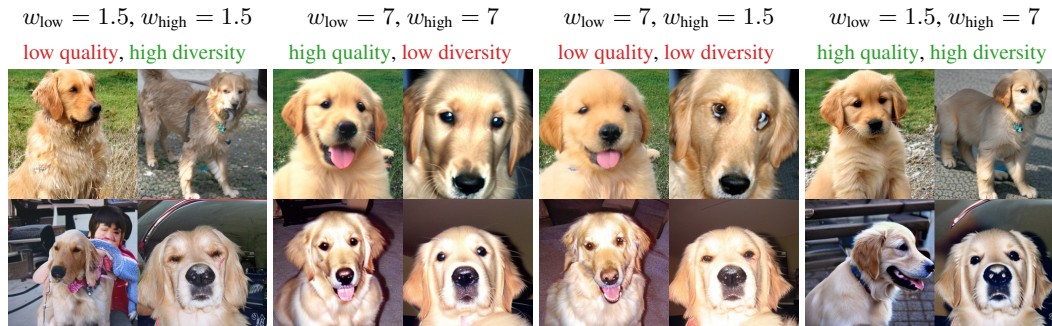

Figure 2: Illustration of how different frequency components of CFG affect generation. Sampling with low CFG scales results in diverse generations but lower overall quality. Increasing the CFG scale improves quality but reduces diversity. We show that the low-frequency component of the CFG signal primarily drives the reduction in diversity, while the high-frequency component contributes to quality enhancement without affecting diversity.

These techniques decompose an image into coarse, low-frequency structures and fine, high-frequency details. The low-frequency components capture the global characteristics of the image, including object placement, overall geometry, and color distribution, while the high-frequency components represent localized information such as edges, textures, and fine structural details.

## 4  CLASSIFIER-FREE GUIDANCE IN THE FREQUENCY DOMAIN

We now describe how guidance in the frequency domain can enhance the characteristics of CFG. Our first goal is to understand how different frequency components in the CFG prediction $\hat{D}_{\text{CFG}}(\boldsymbol{z}_t, t, \boldsymbol{y})$ influence the final generation. Let $\psi[\cdot]$ denote a linear and invertible frequency transformation, such as a Laplacian pyramid or wavelet transform, which decomposes each input $\boldsymbol{x}$ into low- and high-frequency components, denoted by $\psi_{\text{low}}[\boldsymbol{x}]$ and $\psi_{\text{high}}[\boldsymbol{x}]$, respectively. For notational convenience, we define $D_c(\boldsymbol{z}_t) \doteq D_{\boldsymbol{\theta}}(\boldsymbol{z}_t, t, \boldsymbol{y})$, $D_u(\boldsymbol{z}_t) \doteq D_{\boldsymbol{\theta}}(\boldsymbol{z}_t, t, \boldsymbol{y}_{\text{null}})$, and $\hat{D}_{\text{CFG}}(\boldsymbol{z}_t) \doteq \hat{D}_{\text{CFG}}(\boldsymbol{z}_t, t, \boldsymbol{y})$ to represent the conditional, unconditional, and CFG outputs, respectively. As a consequence of the assumed properties of $\psi$, the CFG update rule can be expressed as

$$\hat{D}_{\text{CFG}}(\boldsymbol{z}_t) = \psi^{-1}[\psi[\hat{D}_{\text{CFG}}(\boldsymbol{z}_t)]] \tag{3}$$

$$= \psi^{-1}[\psi[D_u(\boldsymbol{z}_t)] + w(\psi[D_c(\boldsymbol{z}_t)] - \psi[D_u(\boldsymbol{z}_t)])]. \tag{4}$$

This implies that, in the frequency domain, the CFG update affects both the low- and high-frequency components of $\psi[\hat{D}_{\text{CFG}}(\boldsymbol{z}_t)] = \{\hat{D}_{\text{CFG}}^{\text{low}}(\boldsymbol{z}_t), \hat{D}_{\text{CFG}}^{\text{high}}(\boldsymbol{z}_t)\}$ as follows:

$$\hat{D}_{\text{CFG}}^{\text{low}}(\boldsymbol{z}_t) = \psi_{\text{low}}[D_u(\boldsymbol{z}_t)] + w(\psi_{\text{low}}[D_c(\boldsymbol{z}_t)] - \psi_{\text{low}}[D_u(\boldsymbol{z}_t)]), \tag{5}$$

$$\hat{D}_{\text{CFG}}^{\text{high}}(\boldsymbol{z}_t) = \psi_{\text{high}}[D_u(\boldsymbol{z}_t)] + w(\psi_{\text{high}}[D_c(\boldsymbol{z}_t)] - \psi_{\text{high}}[D_u(\boldsymbol{z}_t)]). \tag{6}$$

Thus, in standard CFG, both low- and high-frequency components are guided using the same scale $w$ throughout the sampling process. However, we argue that this approach is suboptimal, as the low- and high-frequency components exhibit different behaviors and influence distinct aspects of the generated image. We find that strong guidance on the low-frequency component $\hat{D}_{\text{CFG}}^{\text{low}}(\boldsymbol{z}_t)$ leads to oversaturation and reduced diversity, whereas high guidance on the high-frequency component $\hat{D}_{\text{CFG}}^{\text{high}}(\boldsymbol{z}_t)$ primarily enhances image quality. On the other hand, low scales for $w_{\text{low}}$ keeps diversity and realistic color composition while low values for $w_{\text{high}}$ degrade the visual details of the image and result in reduced sample quality. Motivated by this, we propose a generalized CFG scheme, called frequency-decoupled guidance (FDG), that employs separate guidance scales—$w_{\text{low}}$ for low-frequency and $w_{\text{high}}$ for high-frequency components.

Figure 2 illustrates the effect of low- and high-frequency components on generated images. For CFG ($w_{\text{low}} = w_{\text{high}}$), low guidance scales lead to poor global structure and visual degradation, while high guidance reduces diversity. We observe that strong guidance on the low-frequency signal (i.e., a large $w_{\text{low}}$) primarily causes diversity issues, whereas increasing $w_{\text{high}}$ enhances quality without adverse effects on diversity. These findings highlight the limitations of using a single scale $w$ in standard CFG: low $w$ produces blurry or incoherent outputs, while high $w$ reduces diversity and causes oversaturation. Our results therefore advocate for asymmetric guidance, where we set $w_{\text{low}} < w_{\text{high}}$.

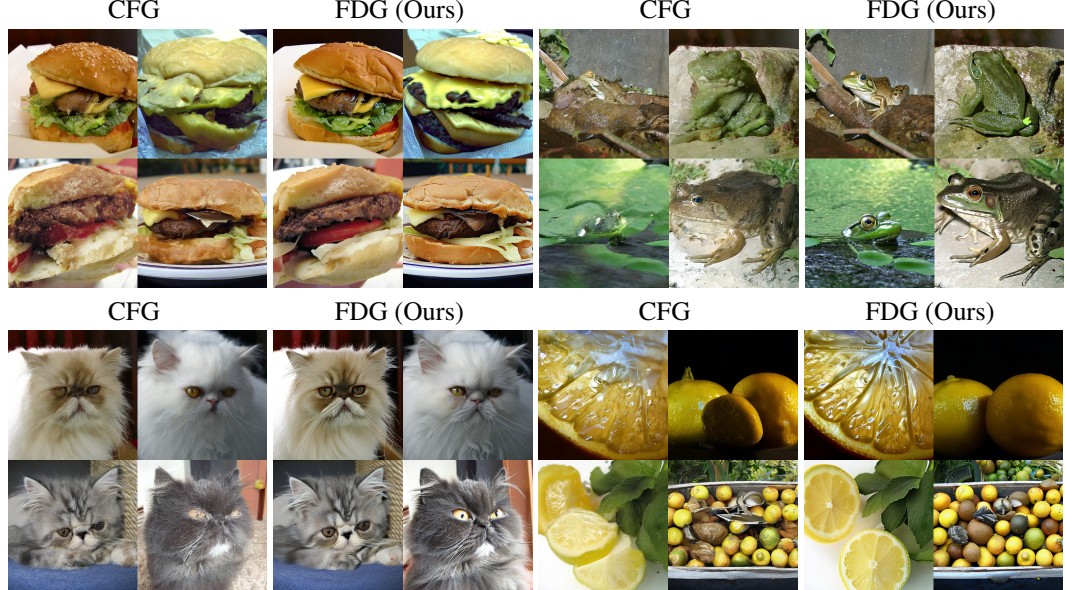

Figure 3: Class-conditional generation results using EDM2 with $w = w_{\text{low}} = 1.25$. FDG enhances image quality while maintaining the diversity of low CFG scales.

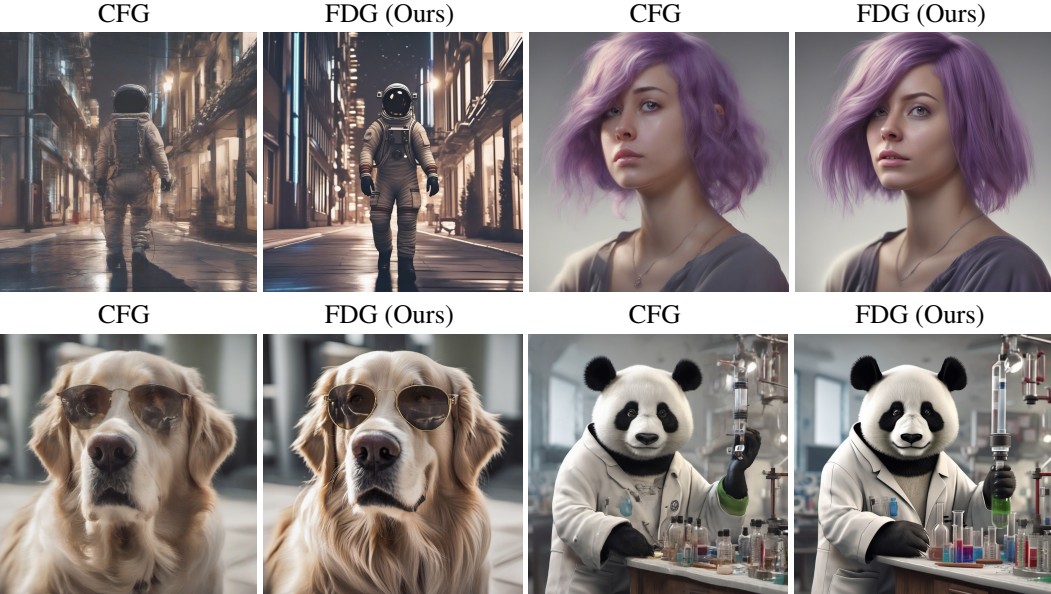

Figure 4: Text-to-image generation results using Stable Diffusion XL with $w = w_{\text{low}} = 2$. FDG enhances the details of the CFG image while maintaining the overall structure and color palette.

**Implementation details**   We use Laplacian pyramids (Burt & Adelson, 1983) as the frequency transform $\psi$ in our experiments. The complete algorithm for applying FDG is given in Algorithm 1, and the pseudocode of our method is provided in Algorithm 2. Notably, FDG requires only minor modifications to the standard CFG sampling procedure, introduces no significant computational overhead, and is readily compatible with all pretrained diffusion models.

## 5   EXPERIMENTS

**Setup**   We primarily conduct experiments on text-to-image generation using Stable Diffusion models (Rombach et al., 2022; Podell et al., 2023; Esser et al., 2024), and class-conditional ImageNet (Russakovsky et al., 2015) generation using EDM2 (Karras et al., 2023) and DiT-XL/2 (Peebles & Xie, 2022). In all cases, we adopt the official pretrained checkpoints and codebases to maintain

Table 1: Quantitative comparison between CFG and FDG. FDG consistently improves FID and recall, showing high generation quality while maintaining good diversity.

| Model | Guidance | FID ↓ | Precision ↑ | Recall ↑ |
|---|---|---|---|---|
| EDM2-S (Karras et al., 2023) | CFG | 9.77 | **0.85** | 0.52 |
| | FDG (Ours) | **5.44** | 0.83 | **0.68** |
| EDM2-XXL (Karras et al., 2023) | CFG | 8.65 | **0.83** | 0.56 |
| | FDG (Ours) | **4.99** | 0.82 | **0.68** |
| DiT-XL/2 (Peebles & Xie, 2022) | CFG | 9.31 | **0.89** | 0.54 |
| | FDG (Ours) | **5.33** | 0.84 | **0.65** |
| Stable Diffusion 2.1 (Rombach et al., 2022) | CFG | 24.99 | 0.68 | 0.44 |
| | FDG (Ours) | **23.33** | **0.69** | **0.49** |
| Stable Diffusion XL (Podell et al., 2023) | CFG | 25.23 | **0.64** | 0.49 |
| | FDG (Ours) | **24.60** | 0.62 | **0.52** |
| Stable Diffusion 3 (Esser et al., 2024) | CFG | 30.32 | **0.76** | 0.37 |
| | FDG (Ours) | **27.68** | **0.76** | **0.42** |

Table 2: Quantitative comparison of CFG and FDG across various evaluation metrics for text-to-image models. FDG consistently outperforms CFG on all metrics and models.

| Benchmark | Model | Guidance | ImageReward ↑ | HPSv2 ↑ | PickScore ↑ | CLIP Score ↑ |
|---|---|---|---|---|---|---|
| DrawBench (Saharia et al., 2022b) | Stable Diffusion 2.1 | CFG | −0.112 | 0.273 | 0.45 | 30.82 |
| | | FDG (Ours) | **0.101** | **0.279** | **0.55** | **31.81** |
| | Stable Diffusion XL | CFG | 0.323 | 0.277 | 0.38 | 31.99 |
| | | FDG (Ours) | **0.595** | **0.286** | **0.62** | **32.78** |
| | Stable Diffusion 3 | CFG | 0.185 | 0.273 | 0.33 | 31.26 |
| | | FDG (Ours) | **0.803** | **0.288** | **0.67** | **33.08** |
| Parti Prompts (Yu et al., 2022) | Stable Diffusion 2.1 | CFG | 0.109 | 0.270 | 0.42 | 31.55 |
| | | FDG (Ours) | **0.332** | **0.277** | **0.58** | **32.09** |
| | Stable Diffusion XL | CFG | 0.442 | 0.276 | 0.48 | 32.22 |
| | | FDG (Ours) | **0.663** | **0.283** | **0.52** | **32.81** |
| | Stable Diffusion 3 | CFG | 0.527 | 0.272 | 0.31 | 31.84 |
| | | FDG (Ours) | **0.988** | **0.286** | **0.69** | **32.83** |
| HPS Prompts (Wu et al., 2023) | Stable Diffusion 2.1 | CFG | 0.035 | 0.269 | 0.35 | 32.73 |
| | | FDG (Ours) | **0.282** | **0.276** | **0.65** | **33.56** |
| | Stable Diffusion XL | CFG | 0.603 | 0.278 | 0.35 | 33.99 |
| | | FDG (Ours) | **0.824** | **0.286** | **0.65** | **34.77** |
| | Stable Diffusion 3 | CFG | 0.470 | 0.272 | 0.29 | 32.22 |
| | | FDG (Ours) | **1.00** | **0.288** | **0.71** | **33.73** |

consistency with the original implementations. We provide extensive additional experiments and thorough ablations in Appendices B and C. Also, more details on the experimental setup and the hyperparameters used for each experiment are given in Appendix D.

**Evaluation metrics** For class-conditional models, we adopt Fréchet Inception Distance (FID) (Heusel et al., 2017) as the main metric to assess both image quality and diversity, given its strong correlation with human perception. To account for FID's sensitivity to implementation details, we evaluate all models under a consistent setup. Additionally, we report precision as a supplementary quality metric and recall to capture diversity (Kynkäänniemi et al., 2019). For text-to-image tasks, we further use ImageReward (Xu et al., 2023), HPSv2 (Wu et al., 2023), PickScore (Kirstain et al., 2023), and CLIP Score (Hessel et al., 2021) to evaluate image quality and prompt alignment.

## 5.1 MAIN RESULTS

**Qualitative results** Figures 3 and 4 qualitatively compare the generations of FDG and CFG for the EDM2 and Stable Diffusion XL models at low CFG scales. FDG enhances generation quality while preserving the overall structure and color palette of the CFG output. Thus, FDG improves the quality at low CFG scales while avoiding the issues caused by high CFG values.

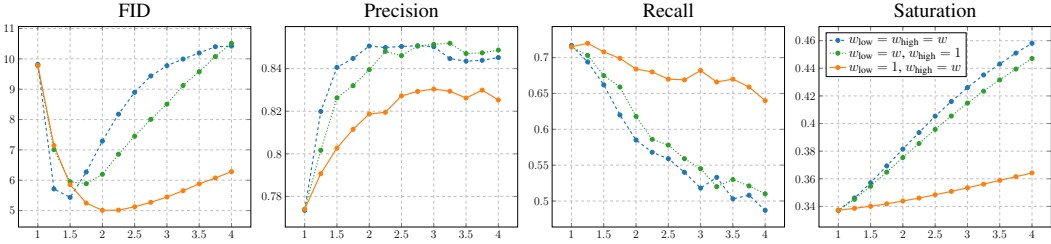

Figure 5: Illustrating the effect of different frequency components on CFG behavior. Although CFG improves quality, it rapidly restricts diversity and increases saturation, leading to higher FID and lower recall. Note that the low-frequency component is mainly responsible for these adverse effects, whereas the high-frequency component enhances quality while preserving diversity and color composition, resulting in better FID and recall.

**Quantitative results** Table 1 provides the metrics for FDG and CFG across several models at guidance values typically used in practice. We observe that FDG consistently improves FID and recall while largely maintaining the precision of the CFG outputs. We therefore conclude that FDG enhances quality while preserving diversity, leading to significantly better FID. Additionally, we evaluate generation quality at lower CFG values in Table 2 using metrics specifically designed to reflect human preferences for text-to-image models. As FDG boosts quality without introducing the detrimental effects of high CFG scales, it significantly outperforms CFG across all quality metrics for the Stable Diffusion models.

## 5.2 Effect of different frequency components on the generated distribution

To directly measure the effect of different frequency components on CFG outputs, we compared CFG with two sampling variants that use only $\hat{D}_{\text{CFG}}^{\text{low}}(z_t)$ or $\hat{D}_{\text{CFG}}^{\text{high}}(z_t)$. This was achieved by setting $w_{\text{high}} = 1$ or $w_{\text{low}} = 1$ in FDG. As shown in Figure 5, increasing $w_{\text{low}}$ is the main cause of the reduced diversity in CFG, as evidenced by the higher FID and lower recall values. In contrast, increasing $w_{\text{high}}$ improves precision while maintaining recall, leading to significantly better FID across most guidance scales. These results suggest that excessive low-frequency guidance is the main contributor to the adverse effects of high CFG, whereas increasing high-frequency guidance generally enhances generation quality. Additionally, we observed that the low-frequency component is the primary cause of oversaturation, explaining why high guidance scales result in color artifacts. Therefore, our method adopts low values for $w_{\text{low}}$ and high values for $w_{\text{high}}$.

## 5.3 Frequency analysis of prompt alignment in CFG

We next demonstrate how different frequency components of the CFG signal influence the alignment between generated images and the input condition. Figure 6 shows that although both low- and high-frequency components improve alignment as the guidance scale increases, the low-frequency component is the primary driver of this effect. Therefore, FDG can achieve comparable or superior prompt alignment to CFG by appropriately combining low- and high-frequency components, as shown in Table 2.

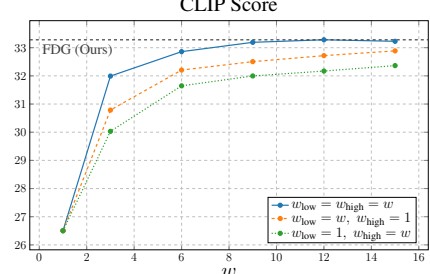

Figure 6: Effect of frequency components of CFG on prompt alignment.

## 5.4 Additional experiments

**Relation to variable guidance scale** Several works have explored the use of time-dependent guidance scales to balance diversity and quality in classifier-free guidance (Sadat et al., 2024a; Kynkäänniemi et al., 2024; WANG et al., 2024). For example, guidance interval (GI) (Kynkäänniemi et al., 2024) applies the guidance scale only during a limited range of sampling steps identified via grid search. We argue that the improvement in diversity offered by these methods is closely related to the frequency decomposition of the guidance signal. To support this, we provide the norms of

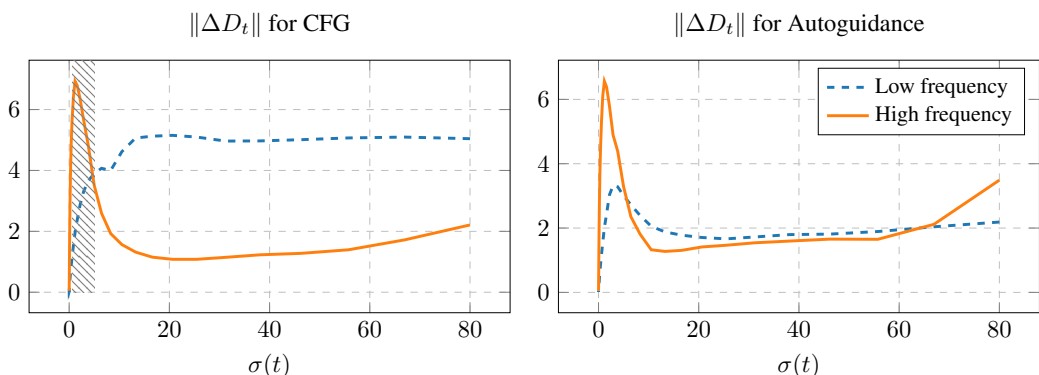

Figure 7: Illustration of how the norm of the guidance signal $\Delta D_t$ behaves in the frequency domain for CFG and Autoguidance. For CFG, low-frequency components are dominant during most early steps (high $\sigma(t)$), which can be harmful. Guidance interval (Kynkäänniemi et al., 2024) improves this by limiting CFG to the shaded region (found by grid search), i.e., the steps where high-frequency components are dominant. In contrast, Autoguidance maintains strong high-frequency norms throughout the sampling process, making the low-frequency signal useful at all steps (Karras et al., 2024).

Table 3: Comparing guidance interval (Kynkäänniemi et al., 2024) with FDG based on Stable Diffusion XL (Podell et al., 2023). FDG achieves better quality metrics due to having high-frequency guidance in earlier sampling steps.

| Method | ImageReward ↑ | HPSv2 ↑ | PickScore ↑ | CLIP Score ↑ |
|---|---|---|---|---|
| Guidance interval | 0.437 | 0.282 | 0.34 | 32.77 |
| FDG (Ours) | **0.595** | **0.286** | **0.66** | **32.78** |

Table 4: Quantitative comparison between CFG and FDG using SDXL-Lightning (Lin et al., 2024) as an example of a distilled model that uses fewer sampling steps. FDG outperforms sampling both with and without CFG, achieving better quality and good prompt alignment.

| Method | ImageReward ↑ | HPSv2 ↑ | CLIP Score ↑ |
|---|---|---|---|
| w/o CFG | 0.535 | 0.282 | 31.77 |
| CFG | 0.573 | 0.286 | **32.44** |
| FDG (Ours) | **0.672** | **0.292** | **32.44** |

the low- and high-frequency components of the guidance signal across sampling steps in Figure 7. Note that the norm of the high-frequency component increases over sampling, while the norm of the low-frequency component decreases. We observe that GI starts applying guidance when the norms of the two components become roughly equal. This suggests that applying guidance in the mid-stages, as GI does, implicitly increases the effective guidance on the high-frequency component relative to the low-frequency component. This analysis also provides a principled way to select an interval for applying GI, avoiding costly grid searches. Additionally, GI may still lead to quality degradation due to the absence of guidance at the beginning of sampling. As shown in Table 3, FDG outperforms GI in terms of image quality for Stable Diffusion XL by maintaining high-frequency guidance during the early sampling steps.

**Frequency analysis of Autoguidance** Autoguidance (Karras et al., 2024) proposed a modified version of CFG that replaces the unconditional prediction with a degraded version of the main diffusion model. Figure 7 presents a frequency analysis of the update provided by Autoguidance, showing that both low- and high-frequency components remain strong throughout the sampling process, unlike CFG where low-frequency response dominates. This likely explains why Autoguidance is effective at all steps, and why it provides a better guidance direction compared to CFG.

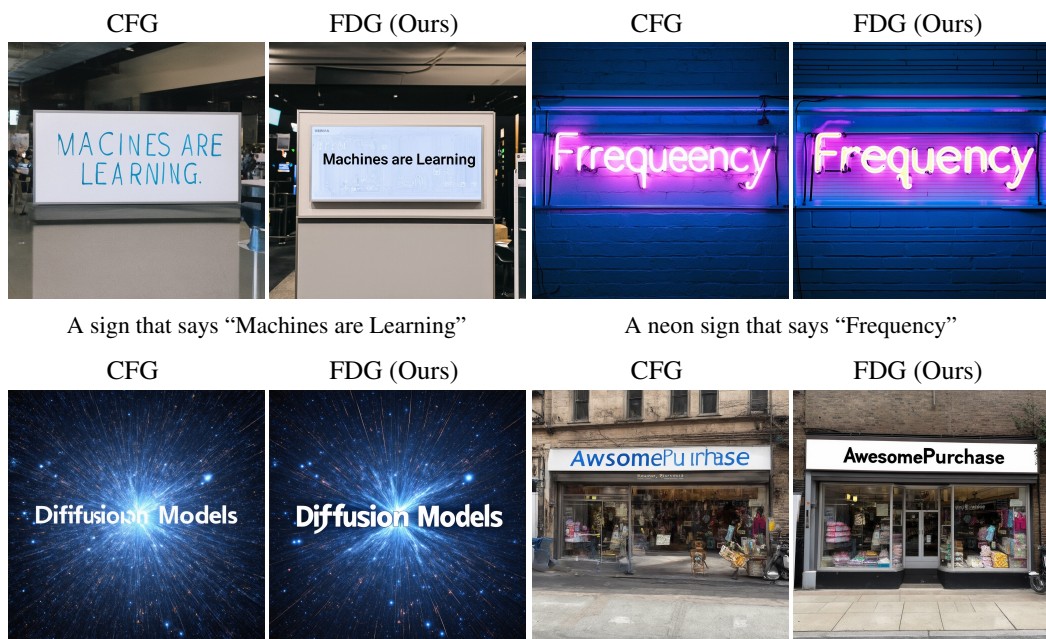

| CFG | FDG (Ours) | CFG | FDG (Ours) |

A sign that says "Machines are Learning"     A neon sign that says "Frequency"

A cosmic sign that says "Diffusion Models"     A storefront with "AwesomePurchase" written on it

Figure 8: Compared with CFG, FDG can improve the quality of text rendering in generated images while also using a low guidance scale to preserve the realism of generated images.

**Compatibility with distilled models** CFG is often detrimental to distilled models that use a small number of sampling steps. In contrast, we show that FDG can be effectively applied to distilled models, such as SDXL-Lightning (Lin et al., 2024), without quality degradation. Table 4 provides a quantitative comparison between FDG and CFG. Compared to both baselines, FDG achieves higher quality metrics with good prompt alignment.

**Improving text rendering in diffusion models** We next demonstrate that FDG can enhance the quality of generated text in Stable Diffusion 3 (Esser et al., 2024). Generating high-quality text requires substantial details, which poses a challenge for standard CFG, since high guidance scales often result in unrealistic samples. In contrast, Figure 8 shows that FDG achieves realistic generation and correct spelling of text by separately controlling $w_{\text{low}}$ and $w_{\text{high}}$.

## 6 CONCLUSION

In this work, we have taken a principled look at classifier-free guidance in the frequency domain and have shown that its beneficial effects on structural fidelity and fine details stem from strong guidance applied to the high-frequency components of the CFG signal, while its detrimental impact on diversity and oversaturation arises from excessive guidance on the low-frequency components. Building on this insight, we proposed frequency-decoupled guidance (FDG) to disentangle guidance strength across frequency bands, applying conservative scaling to low frequencies while exploiting stronger scaling at high frequencies. This approach preserves the diversity and color composition of low guidance scales while enhancing details akin to high guidance scales. As a result, FDG improves the quality of low CFG values while avoiding the adverse effects of high CFG scales by design. Importantly, FDG introduces practically no additional training or sampling cost and can be seamlessly integrated as a *plug-and-play* enhancement to any pretrained diffusion model using CFG. As with CFG itself, challenges remain in accelerating sampling and improving generation quality in extreme out-of-distribution domains, which we identify as promising directions for future research.

BROADER IMPACT STATEMENT

Our approach has the potential to enhance the realism and quality of outputs generated by diffusion models without the need for costly retraining. As such, it offers practical benefits for visual content creation. However, with the continued advancement of generative modeling, the generation and dissemination of fabricated or inaccurate data become increasingly accessible. While advancements in AI-generated content have the potential to enhance productivity and creativity, it remains essential to critically assess the accompanying risks and ethical considerations. For a comprehensive discussion on ethics and creativity within computer vision, we direct readers to Rostamzadeh et al. (2021).

REPRODUCIBILITY STATEMENT

Our work leverages the official implementations of the pretrained models referenced in the main text. The inference procedure for applying FDG is outlined in Algorithm 1, while the corresponding pseudocode is provided in Algorithm 2. Further implementation details, including the hyperparameters used in our main experiments, are described in Appendix D.

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

## A    BACKGROUND ON FREQUENCY DECOMPOSITIONS

**Laplacian Pyramids**   The Laplacian pyramid (Burt & Adelson, 1983) is a multi-scale representation of an image based on successive band-pass filtering. Starting from an input image $x$, a Gaussian pyramid $\{G_0, G_1, \ldots, G_N\}$ is constructed, where $G_0 = x$, and each level is a progressively downsampled (and low-pass filtered) version of the previous:

$$G_{i+1} = \texttt{Downsample}(\texttt{GaussianBlur}(G_i)). \tag{7}$$

The Laplacian pyramid $\{L_0, L_1, \ldots, L_{N-1}\}$ is then formed by subtracting the upsampled version of each Gaussian level from its corresponding higher-resolution level:

$$L_i = G_i - \texttt{Upsample}(G_{i+1}). \tag{8}$$

The top level of the pyramid is typically the final low-resolution image $G_N$. The decomposition can be inverted to reconstruct the original signal by sequentially upsampling and summing the Laplacian levels:

$$G_i = L_i + \texttt{Upsample}(G_{i+1}). \tag{9}$$

This approach enables localized manipulation of image details at different spatial scales and is commonly used in image compression, enhancement, and blending (Szeliski, 2022).

**Wavelet transforms**   Wavelet transforms (Brewster, 1993) are widely used in signal processing to extract spatial-frequency characteristics from input data. They rely on a pair of filters: a low-pass filter $L$ and a high-pass filter $H$. In the case of 2D signals, four filters are derived as $LL^\top$, $LH^\top$, $HL^\top$, and $HH^\top$. When applied to an image $x$, the 2D wavelet transform decomposes it into a low-frequency sub-band $x_L$ and three high-frequency sub-bands $\{x_H, x_V, x_D\}$, which capture horizontal, vertical, and diagonal details, respectively. For an input image of size $H \times W$, each sub-band has a spatial size of $H/2 \times W/2$. Multi-resolution analysis can be performed by recursively applying the wavelet transform to $x_L$. The transform is also invertible, allowing the reconstruction of the original image $x$ from the set $\{x_L, x_H, x_V, x_D\}$ using the inverse wavelet transform. Moreover, fast wavelet transform (FWT) (Mallat, 1989) makes it possible to compute wavelet sub-bands with linear complexity relative to the number of pixels in $x$.

## B    ADDITIONAL EXPERIMENTS

We present additional experiments in this section to further demonstrate the effectiveness of FDG across various scenarios and its compatibility with several alternative guidance methods.

### B.1    EFFECT OF LOW- AND HIGH-FREQUENCY COMPONENTS ON GENERATIONS

To further demonstrate that low-frequency components govern global structure and high-frequency components contribute to details, we conducted an experiment in which we explicitly set either the high- or low-frequency portion of the CFG signal to zero. The results in Figure 9 show that when high frequencies are removed, the final generation retains the overall structure of the base image. Conversely, when low frequencies are removed, the generated image roughly shows the high-frequency details of the base image such as edges. These findings suggest that low-frequency components in the CFG signal control global structure, while high-frequency components determine more localized details.

### B.2    COMPATIBILITY WITH ADAPTIVE PROJECTED GUIDANCE

Adaptive projected guidance (APG) (Sadat et al., 2025) is a method designed to reduce oversaturation and artifacts caused by high guidance scales. APG complements our methods, as its update rule can be integrated with the frequency decomposition in FDG to get better guidance directions. We demonstrate this combined approach in Figure 10 by incorporating the orthogonal projection from APG into FDG. The results indicate that this projection continues to effectively produce more realistic color compositions and fewer artifacts in generations.

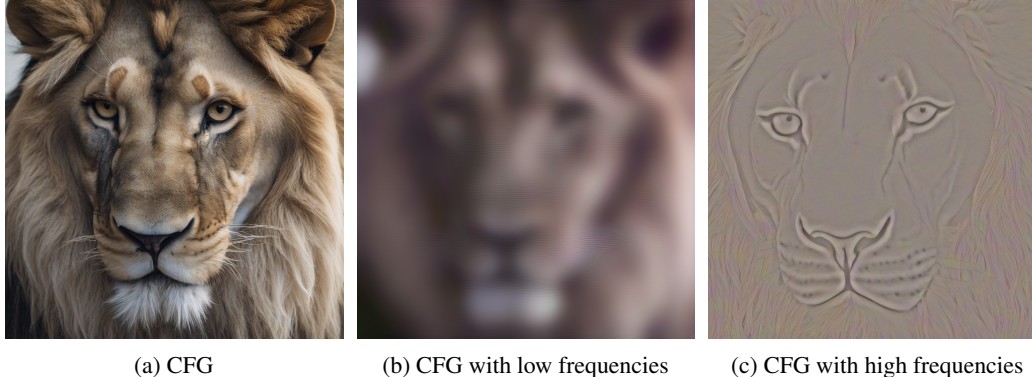

| (a) CFG | (b) CFG with low frequencies | (c) CFG with high frequencies |

Figure 9: Effect of low- and high-frequency components of CFG on generated results. In this experiment, we explicitly set either low or high frequencies to zero to isolate the effect of the other component. Note that low frequencies determine the overall structure of the output, while high frequencies contribute to details.

FDG    FDG + APG    FDG    FDG + APG

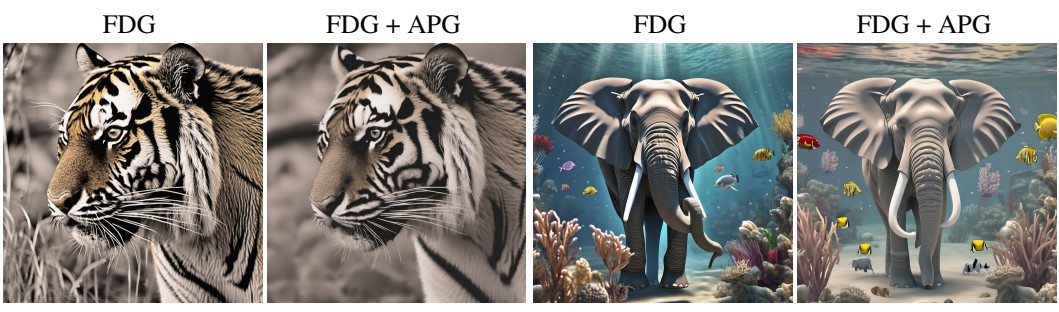

Image of a tiger    Elephant under the see, realistic, 4k

Figure 10: Combining FDG with APG. The APG can be seamlessly integrated with FDG to produce generations with more realistic colors. For this experiment, we set $w_{\text{low}} = 5$ and $w_{\text{high}} = 15$.

## B.3 CHANGING THE WEIGHTS IN FDG

In this part, we further evaluate the effect of varying $w_{\text{low}}$ and $w_{\text{high}}$ on generation quality based on the EDM2-S model. For use $w_{\text{high}} \in \{1.5, 2, 2.5\}$, and we set $w_{\text{low}} = r(w_{\text{high}} - 1) + 1$ with $r \in \{0, 0.25, 0.5, 0.75, 1\}$. Figure 11 shows that across all settings, increasing $w_{\text{high}}$ and reducing the strength of the low-frequency component ($r < 1$) improves FID and recall while maintaining a comparable level of precision. Based on this, we recommend setting $w_{\text{high}}$ relatively high and choosing $r \leq 0.5$ across models. This experiment further demonstrates that by lowering the weight of the low-frequency component, FDG can simultaneously capture the benefits of both low and high CFG scales, i.e., achieving high precision and recall along with low FID.

## B.4 COMBINING FDG WITH CADS

CADS (Sadat et al., 2024a) is an inference method designed to increase the diversity of diffusion models at high guidance scales by perturbing the conditional embedding with Gaussian noise. In this section, we show that CADS is compatible with our method, and that their combination outperforms either approach used in isolation. Table 5 supports this finding using the DiT-XL/2 model as a benchmark. We therefore conclude that the benefits of FDG are complementary to those of CADS.

## B.5 USING DIFFERENT DIFFUSION SAMPLERS

We also show that the effectiveness of FDG is not limited to a specific diffusion sampler. Table 6 compares the metrics of FDG and CFG across several popular diffusion samplers for DiT-XL/2, demonstrating that FDG consistently outperforms CFG across all setups.

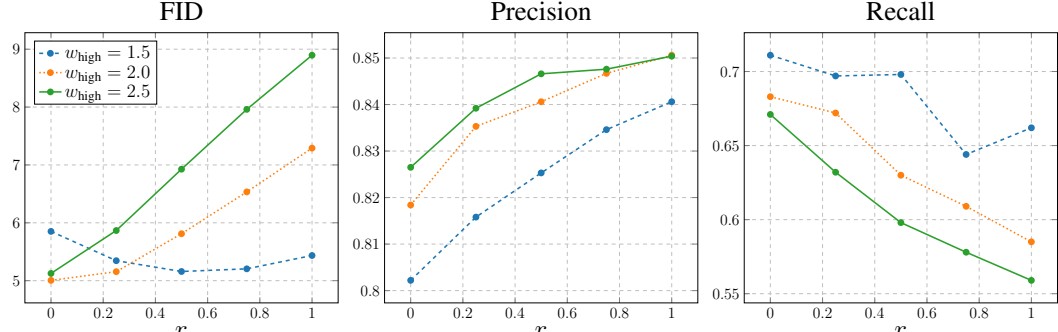

Figure 11: Quantitative evaluation of varying $w_{\text{low}}$ relative to $w_{\text{high}}$ (defined as $w_{\text{low}} = r(w_{\text{high}}-1)+1$). Compared to the CFG baseline ($r = 1$), increasing the high-frequency weight ($w_{\text{high}}$) while reducing the low-frequency weight ($r < 1$) consistently improves FID and recall while maintaining precision. The best trade-off is achieved when $w_{\text{high}}$ is relatively high and $r < 0.5$.

Table 5: Effectiveness of CADS on FDG using DiT-XL/2 at a guidance scale of 5. Combining FDG with CADS yields the best FID, outperforming each method used in isolation.

| Guidance | FID ↓ | Precision ↑ | Recall ↑ |
|---|---|---|---|
| CFG | 21.48 | **0.92** | 0.30 |
| CFG + CADS | 14.53 | 0.88 | 0.48 |
| FDG (Ours) | 13.90 | 0.90 | 0.48 |
| FDG + CADS (Ours) | **8.98** | 0.81 | **0.61** |

Table 6: Impact of applying FDG with popular diffusion samplers on the class-conditional ImageNet model (DiT-XL/2). FDG achieves improved FID and recall across all samplers.

| | FDG (Ours) | | CFG | |
|---|---|---|---|---|
| Sampler | FID ↓ | Recall ↑ | FID ↓ | Recall ↑ |
| DDIM (Song et al., 2021a) | **4.84** | **0.69** | 6.91 | 0.60 |
| DPM++ (Lu et al., 2022b) | **4.77** | **0.69** | 7.11 | 0.61 |
| SDE-DPM++ (Lu et al., 2022b) | **5.06** | **0.68** | 9.10 | 0.56 |
| PNDM (Liu et al., 2022a) | **4.75** | **0.69** | 7.02 | 0.61 |
| UniPC (Zhao et al., 2023) | **4.76** | **0.69** | 7.16 | 0.60 |

### B.6 CHANGING THE NUMBER OF SAMPLING STEPS

We also evaluated the performance of FDG and CFG across different numbers of sampling steps. Figure 12 shows that FDG maintains a consistent advantage over CFG across various sampling budgets, leading to improved FID and recall while preserving a similar level of precision. Therefore, we conclude that the observed improvements in FDG hold across different sampling budgets.

### B.7 COMPATIBILITY WITH OTHER GUIDANCE METHODS

While our approach primarily focuses on beneficial weighting of frequency components tailored to the unique dynamics of CFG, the underlying principle of decoupling these components during guidance is more broadly applicable. FDG can be naturally extended to other guidance methods, though the optimal weighting schemes may differ depending on the specific characteristics of each algorithm. To investigate this generalization, we conducted additional experiments applying frequency-specific scaling to two alternative guidance strategies: perturbed attention guidance (PAG) (Ahn et al., 2024) with Stable Diffusion XL and autoguidance (Karras et al., 2024) with EDM2. The corresponding results in Tables 7 and 8 highlight the impact of FDG beyond CFG.

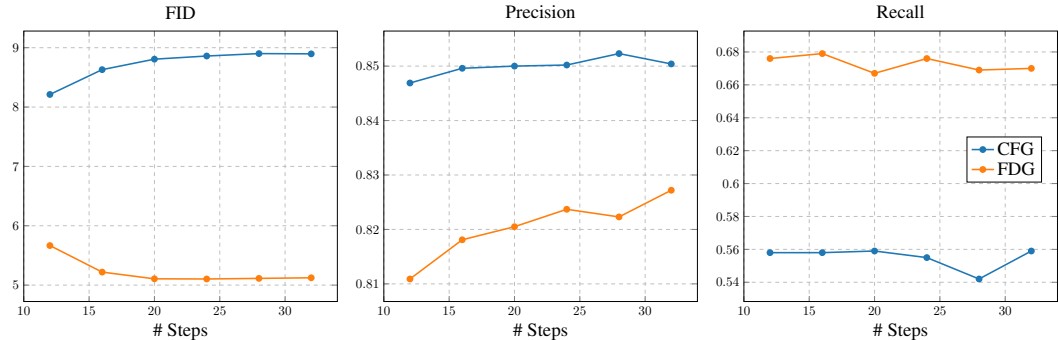

Figure 12: Comparison between FDG and CFG across different numbers of sampling steps. FDG consistently outperforms CFG, maintaining a clear FID improvement at all sampling budgets.

Table 7: Comparison of PAG and PAG+FDG with different weighting configurations.

| Method | ImageReward ↑ | HPSv2 ↑ | HPS Win Rate ↑ | CLIP Score ↑ |
|---|---|---|---|---|
| PAG ($w = 4$) | -0.13 | 0.270 | 0.34 | 29.28 |
| +FDG ($w_{low} = 4$, $w_{high} = 7$) | -0.05 | 0.272 | 0.63 | **29.33** |
| +FDG ($w_{low} = 7$, $w_{high} = 4$) | **-0.03** | **0.273** | **0.66** | **29.33** |

Table 8: Comparison of Autoguidance and FDG on EDM2 with different weighting schemes.

| Method | FID ↓ | Precision ↑ | Recall ↑ |
|---|---|---|---|
| autoguidance ($w = 3$) | 4.82 | 0.77 | **0.73** |
| +FDG ($w_{low} = 3$, $w_{high} = 1.5$) | **4.63** | **0.79** | **0.73** |

Table 9: Comparison of constant vs. dynamic weighting for FDG based on EDM2-S.

| Method | FID ↓ | Precision ↑ | Recall ↑ |
|---|---|---|---|
| CFG ($w = 2.5$) | 7.45 | 0.84 | 0.57 |
| FDG (constant) | **5.12** | **0.83** | **0.67** |
| FDG (dynamic) | 5.23 | **0.83** | 0.66 |

Interestingly, while the standard FDG configuration improves the performance of PAG, we observed that PAG exhibits sampling dynamics that differ from those of CFG. In particular, low-frequency components are underrepresented relative to the norm of the update, whereas in CFG the low-frequency components tend to dominate. Consequently, applying stronger guidance to low frequencies relative to high frequencies proved beneficial for PAG. A similar trend was observed with autoguidance, where performance improved with $w_{low} > w_{high}$. This is consistent with the analysis in Figure 7, which illustrates that autoguidance dynamics fundamentally differ from CFG by exhibiting stronger high-frequency components in the update steps. Overall, these results demonstrate that employing distinct guidance scales for low- and high-frequency components (i.e., $w_{low} \neq w_{high}$) is beneficial across both PAG and autoguidance.

### B.8 ADAPTIVE WEIGHTS FOR FDG

Our pipeline supports time-dependent values for $w_{low}$ and $w_{high}$. While the main experiments used constant scaling for simplicity, adaptive weighting of these parameters is also possible. To illustrate this, we conducted an experiment with EDM2-S using time-dependent weighting for FDG. For the dynamic variant, we set $w_{low} = 1 + 0.5(1 - t)$ and $w_{high} = 1 + 1.5(1 - t)$, where $t \in [0, 1]$. Table 9 shows that both FDG variants outperform CFG, and the constant-weight configuration achieves a lower FID. As such, we opted for the constant weighting strategy in our main experiments.

Table 10: Effect of combining FDG with FreeU based on Stable Diffusion 2.1.

| Configuration | Image Reward ↑ | HPSv2 ↑ | CLIP Score ↑ |
|---|---|---|---|
| CFG | -0.112 | 0.273 | 30.82 |
| CFG + FreeU | -0.043 | 0.274 | 30.84 |
| FDG | 0.101 | 0.280 | **31.81** |
| FDG + FreeU | **0.158** | **0.280** | 31.64 |

Table 11: Ablation of the frequency decomposition operator $\psi$ based on the EDM2 model.

(a) The choice of the frequency decomposition function

| $\psi$ | FID ↓ | Precision ↑ | Recall ↑ |
|---|---|---|---|
| CFG | 8.89 | **0.85** | 0.55 |
| Laplacian pyramid | **5.12** | 0.83 | 0.67 |
| Wavelet transform | 5.26 | 0.81 | **0.71** |

(b) multi-level vs single-level transformation

| Config | FID ↓ | Precision ↑ | Recall ↑ |
|---|---|---|---|
| CFG | 8.89 | **0.85** | 0.55 |
| single-level | **5.12** | 0.83 | **0.67** |
| multi-level | 5.25 | 0.83 | **0.67** |

### B.9 COMPLEMENTARITY WITH FREEU

FreeU (Si et al., 2024) is a method designed to improve the denoising quality of UNet-based diffusion models and is complementary to classifier-free guidance (CFG). Since FreeU operates on top of CFG, we can directly replace CFG with FDG in FreeU. Table 10 confirms that the effects of the two methods are indeed complementary. Notably, FDG alone outperforms both CFG and CFG + FreeU, while FDG + FreeU achieves the highest overall quality. These findings demonstrate that FDG and FreeU address complementary aspects of the generation process, distinguishing our contribution from Si et al. (2024). It is also worth noting that FreeU is applicable only to UNet-based models, whereas FDG can be applied to any architecture.

## C ABLATION STUDIES

**Effect of the frequency decomposition operator**  We next test the performance of FDG for two choices of $\psi$ in Table 11a. We note that FDG is not sensitive to this design choice as long as the operation provides an informative low- and high-frequency component. We chose the Laplacian pyramid, as it slightly outperformed DWT in our experiments.

**Using multi-level frequency decomposition**  We also experimented with a multi-level Laplacian pyramid for frequency decomposition, applying different guidance scales to each frequency level. Table 11b shows both multi-level and single-level approaches outperform CFG. For simplicity, our main experiments used a single-level pyramid, though multi-level decomposition can help control separate high-frequency bands and remains viable for applications needing such control.

**Sampling time**  On Stable Diffusion 3 (Esser et al., 2024), both CFG and FDG achieve a throughput of 1.22 iterations per second with identical memory usage when sampling with a batch size of four. This result clearly demonstrates that the frequency operations are lightweight and introduce virtually no additional overhead to the sampling process.

## D IMPLEMENTATION DETAILS

The sampling algorithm of FDG is provided in Algorithm 1, and the corresponding PyTorch implementation is given in Algorithm 2. Compared to CFG, FDG only adds a few extra lines of code and does not incur any noticeable computational overhead. As stated in the code, we convert the model's output to the denoised estimate $D_{\boldsymbol{\theta}}(\boldsymbol{z}_t, t, \boldsymbol{y})$ (also known as the $x_0$ prediction), apply the guidance step, and then convert it back to the original output format at each sampling step. For latent diffusion models, the frequency decomposition is performed in the latent space. The guidance scales are selected in the same way practitioners typically choose CFG values for a model, i.e., generating a few samples and visually inspecting them. This is standard practice in diffusion model use, requires

Table 12: Guidane parameters used to compare the performance of FDG with CFG.

(a) Guidance parameters used for Table 1.

| Model | $w$ | $w_{\text{low}}$ | $w_{\text{high}}$ |
|---|---|---|---|
| EDM2-S | 3 | 1 | 3 |
| EDM2-XL | 2 | 1 | 2 |
| DiT-XL/2 | 2 | 1 | 2 |
| Stable Diffusion 2.1 | 7 | 3 | 7 |
| Stable Diffusion XL | 10 | 5 | 10 |
| Stable Diffusion 3 | 7 | 3 | 7 |

(b) Guidance parameters used for Table 2.

| Model | $w$ | $w_{\text{low}}$ | $w_{\text{high}}$ |
|---|---|---|---|
| Stable Diffusion 2.1 | 3 | 3 | 12 |
| Stable Diffusion XL | 3 | 3 | 12 |
| Stable Diffusion 3 | 1.5 | 1.5 | 12 |

---

**Algorithm 1** Guided sampling with FDG

---

**Require:** Frequency decomposition operators $\psi[\cdot]$ and $\psi^{-1}[\cdot]$ (e.g., Laplacian pyramid)
**Require:** Guidance weights $w_{\text{low}}$ (low-frequency), $w_{\text{high}}$ (high-frequency)
**Require:** Conditioning input $\boldsymbol{y}$
1: Initialize: $\boldsymbol{z}_1 \sim \mathcal{N}(\mathbf{0}, \sigma_{\max}^2 \boldsymbol{I})$
2: **for** $t = \{1, 1 - \delta t, \ldots, 0\}$ **do**
3:     Compute the frequency decomposition of the conditional and unconditional predictions:

$$\psi[D_c(\boldsymbol{z}_t)] = \{\psi_{\text{low}}[D_c(\boldsymbol{z}_t)], \psi_{\text{high}}[D_c(\boldsymbol{z}_t)]\}$$

$$\psi[D_u(\boldsymbol{z}_t)] = \{\psi_{\text{low}}[D_u(\boldsymbol{z}_t)], \psi_{\text{high}}[D_u(\boldsymbol{z}_t)]\}$$

4:     Compute the low- and high-frequency components of FDG

$$\hat{D}_{\text{FDG}}^{\text{low}}(\boldsymbol{z}_t) = \psi_{\text{low}}[D_u(\boldsymbol{z}_t)] + w_{\text{low}}(\psi_{\text{low}}[D_c(\boldsymbol{z}_t)] - \psi_{\text{low}}[D_u(\boldsymbol{z}_t)])$$

$$\hat{D}_{\text{FDG}}^{\text{high}}(\boldsymbol{z}_t) = \psi_{\text{high}}[D_u(\boldsymbol{z}_t)] + w_{\text{high}}(\psi_{\text{high}}[D_c(\boldsymbol{z}_t)] - \psi_{\text{high}}[D_u(\boldsymbol{z}_t)])$$

5:     Convert the guided prediction to the data space using the inverse transform:

$$\hat{D}_{\text{FDG}}(\boldsymbol{z}_t) = \psi^{-1}[\{\hat{D}_{\text{FDG}}^{\text{low}}(\boldsymbol{z}_t), \hat{D}_{\text{FDG}}^{\text{high}}(\boldsymbol{z}_t)\}]$$

6:     Perform one sampling step (e.g., one step of DDIM):

$$\boldsymbol{z}_{t-1} = \text{diffusion\_reverse}(\hat{D}_{\text{FDG}}, \boldsymbol{z}_t, t)$$

7: **end for**
8: **return** $\boldsymbol{z}_0$

---

no retraining, and incurs negligible computational cost. Moreover, as shown in Figure 5, FDG outperforms CFG across a wide range of parameter values, demonstrating robustness to scale choice.

For evaluation, we mainly rely on the ADM evaluation suite (Dhariwal & Nichol, 2021) to calculate FID, precision, and recall. For class-conditional ImageNet models, FID is computed using 10,000 generated images along with the complete training dataset. In the case of text-to-image models, FID is measured using the validation split of MS COCO 2017 (Lin et al., 2014). To evaluate text-to-image quality metrics such as ImageReward (Xu et al., 2023), we follow the official implementations and use the authors' provided test datasets. For PickScore (Kirstain et al., 2023), we calculate the win probability and report a win if one image outperforms the other by a margin greater than 0.1; otherwise, the result is reported as a tie. Details of the hyperparameters used in our main experiments are listed in Table 12.

## E  MORE VISUAL EXAMPLES

This section provides additional samples comparing the performance of FDG with CFG. Figure 13 presents further results for class-conditional generation using DiT-XL/2 and EDM2. For both models, we observe that FDG preserves the structure of the base image while significantly enhancing the details. Additional results for text-to-image models are shown in Figures 14 and 15.

**Algorithm 2** PyTorch implementation of FDG.

```python
import torch
import kornia
from kornia.geometry.transform import build_laplacian_pyramid

def project(
    v0: torch.Tensor, # [B, C, H, W]
    v1: torch.Tensor, # [B, C, H, W]
):
    dtype = v0.dtype
    v0, v1 = v0.double(), v1.double()
    v1 = torch.nn.functional.normalize(v1, dim=[-1, -2, -3])
    v0_parallel = (v0 * v1).sum(dim=[-1, -2, -3], keepdim=True) * v1
    v0_orthogonal = v0 - v0_parallel
    return v0_parallel.to(dtype), v0_orthogonal.to(dtype)

def build_image_from_pyramid(pyramid):
    img = pyramid[-1]
    for i in range(len(pyramid) - 2, -1, -1):
        img = kornia.geometry.pyrup(img) + pyramid[i]
    return img

# We assume all model predictions are converted to "x_0" prediction.
def laplacian_guidance(
    pred_cond: torch.Tensor,   # [B, C, H, W]
    pred_uncond: torch.Tensor, # [B, C, H, W]
    guidance_scale=[1.0, 1.0], # Guidance scales from high- to low-frequency
    parallel_weights=None,     # Optional weights for projection
):
    levels = len(guidance_scale)
    if parallel_weights = None:
        parallel_weights = [1.0] * levels

    pred_cond_pyramid = build_laplacian_pyramid(pred_cond, levels)
    pred_uncond_pyramid = build_laplacian_pyramid(pred_uncond, levels)

    pred_guided_pyramid = []
    parameters = zip(
        pred_cond_pyramid, pred_uncond_pyramid, guidance_scale, parallel_weights
        )
    for idx, (p_cond, p_uncond, scale, par_weight) in enumerate(parameters):
        diff = p_cond - p_uncond
        diff_parallel, diff_orthogonal = project(diff, p_cond)
        diff = par_weight * diff_parallel + diff_orthogonal
        p_guided = p_cond + (scale - 1) * diff
        pred_guided_pyramid.append(p_guided)
    pred_guided = build_image_from_pyramid(pred_guided_pyramid)
    return pred_guided.to(pred_cond.dtype)
```

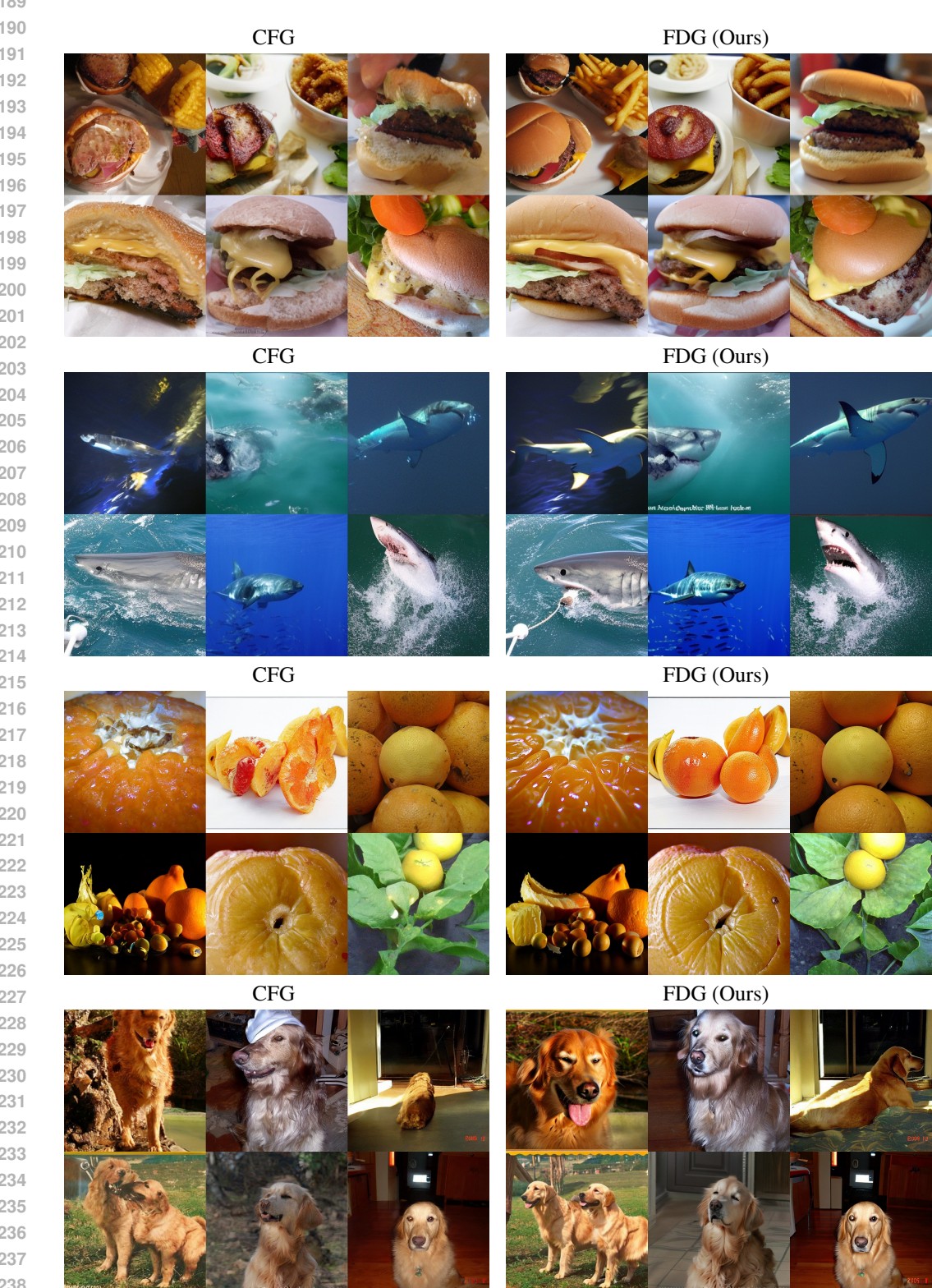

Figure 13: More visual results comparing FDG and CFG using class-conditional ImageNet models.

CFG      FDG (Ours)      CFG      FDG (Ours)

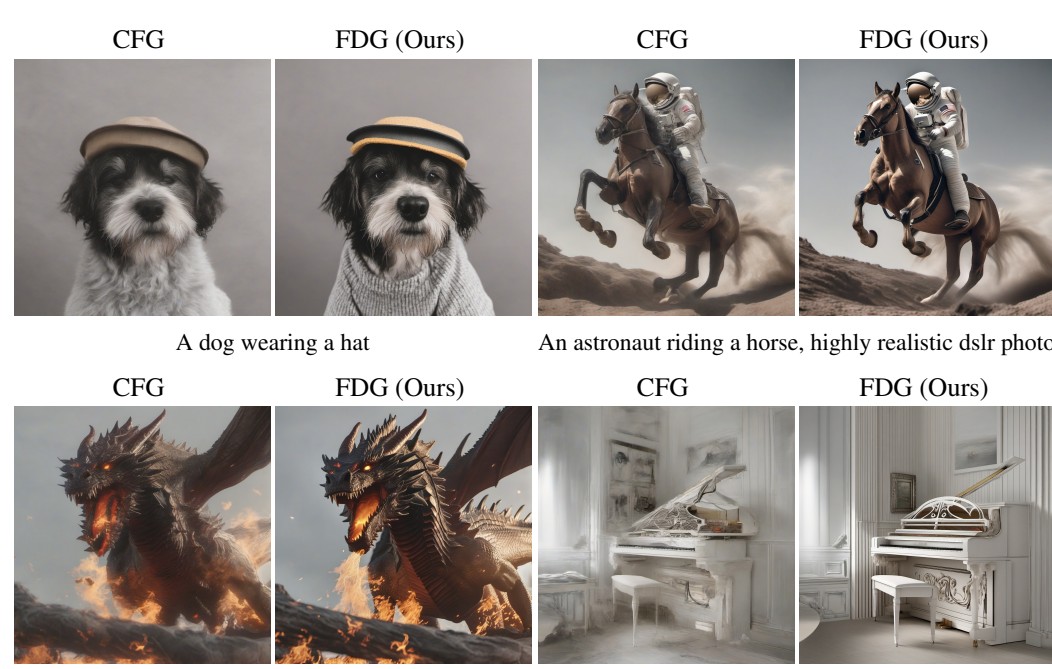

A dog wearing a hat      An astronaut riding a horse, highly realistic dslr photo

CFG      FDG (Ours)      CFG      FDG (Ours)

a close-up of a fire spitting dragon, cinematic shot.      A white piano

Figure 14: More visual examples comparing FDG with CFG using Stable Diffusion XL.

CFG      FDG (Ours)      CFG      FDG (Ours)

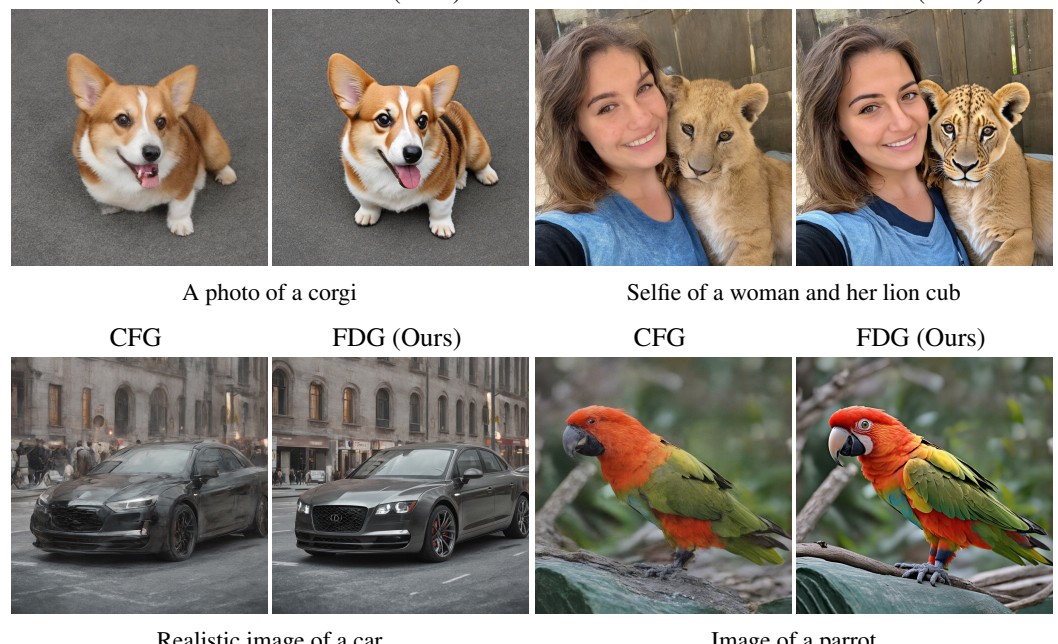

A photo of a corgi      Selfie of a woman and her lion cub

CFG      FDG (Ours)      CFG      FDG (Ours)

Realistic image of a car      Image of a parrot

Figure 15: More visual examples comparing FDG with CFG using Stable Diffusion 3.

