# OpenReview forum: "Guidance in the Frequency Domain Enables High-Fidelity Sampling at Low CFG Scales"
_ICLR.cc/2026/Conference — Submitted to ICLR 2026_

### Official Review · Reviewer_qzZU · 2025-10-27

**Soundness:** 2
**Presentation:** 1
**Contribution:** 2
**Rating:** 2
**Confidence:** 3

**Summary:**

This paper analyzes the role of Classifier-Free Guidance (CFG) in conditional diffusion models, specifically examining its differential impact on low- and high-frequency components during image generation. The authors posit that an excessively high CFG scale applied to the low-frequency domain results in color oversaturation and reduced sample diversity.

To address this, the paper introduces Frequency Decoupled Guidance (FDG), a novel method that utilizes the invertible and linear Laplacian pyramid transformation to decompose the image into distinct frequency bands. FDG then applies different guidance scales to these components—notably, a lower scale for low-frequencies and a higher scale for high-frequencies. The efficacy of FDG is evaluated against standard CFG across various class-conditional and text-to-image generation models.

**Strengths:**

1. This work provides an initial analysis that distinguishes the functional roles of the low- and high-frequency components within the CFG mechanism.

2. The proposed FDG method is training-free. It can be readily integrated into any pre-trained conditional diffusion model with minimal implementation overhead.

**Weaknesses:**

1. Insufficient Evidence: The central claim regarding the distinct roles of CFG's low- and high-frequency components is insufficiently substantiated. It appears to rest primarily on a few qualitative samples presented in Figure 2, which is not robust enough to support such a definitive conclusion.

2. Lack of Clarity: The paper suffers from a significant lack of clarity regarding critical experimental settings.
- The resolution of the ImageNet models used (e.g., for EDM) is not specified, though it is presumed to be $512 \times 512$.
- For a paper centered on guidance, the omission of specific guidance scales in the main text is a critical flaw. Key information is missing: The FDG scales ($w_{low}$, $w_{high}$) used for the visualization in Figure 1.The corresponding $w_{high}$ value used in Figures 3 and 4.
- The guidance scales used to generate the quantitative results in Tables 1 and 2. While this information is reportedly located in the supplementary material, its absence from the main body severely hinders readability and prevents a clear assessment of the paper's claims.

3. Unfair Comparison: A valid comparison between CFG and FDG necessitates evaluating both methods at their respective optimal guidance scales. However, the scales selected for the CFG baselines in Tables 1 (EDM) and 2 (Stable Diffusion) do not appear to align with those from official repositories or standard evaluation practices. The qualitative examples also seem to use suboptimal scales for CFG, potentially skewing the comparison.

4. Inconsistent Performance: The reported baseline performance metrics are inconsistent with established benchmarks. For instance, the official EDM2-S and EDM2-XXL models achieve FIDs of 2.23 and 1.81 (scales 1.40/1.20) and FIDs of 2.56 and 1.91 (even without CFG), respectively. This paper, however, reports much poorer CFG baseline FIDs of 9.77 and 8.65. Any deviations in the evaluation protocol (e.g., the number of images used for FID calculation) that might explain this significant discrepancy are not described.

5. Potential Cherry-Picking of Parameters: The use of different guidance scales for the Stable Diffusion model when reporting metrics in Table 1 versus Table 2 is highly suspect. This suggests that the scales may have been selected via parameter search to find specific, narrow settings where FDG artifactually outperforms CFG on a given metric, rather than demonstrating robust and general superiority.

6. Omission of Critical Metrics: The paper omits the Inception Score (IS). IS is generally more sensitive to sample fidelity, whereas FID is more sensitive to diversity. It is a known phenomenon that FID can often be improved at the expense of IS. The provided precision (lower) and recall (higher) metrics already suggest such a trade-off (sacrificing fidelity for diversity). Without the IS metric, it is impossible to verify if FDG genuinely improves generation quality or simply trades fidelity for a better FID score.

7. Missing Comparison and Overclaiming: The paper fails to compare against relevant prior art, most notably CFG++. Consequently, the claim that this work is the first approach to improve image quality at low CFG scales is an overstatement, as CFG++ addresses a very similar problem.

**Questions:**

1. Could you provide a more rigorous validation of the distinct roles of low/high-frequency components beyond the few samples in Figure 2? Furthermore, to improve clarity, could you please state the exact guidance scales (e.g., $w_{low}$, $w_{high}$, and baseline $w$) used for all figures and tables directly in the main paper?

2. The reported baseline FID scores for EDM models (e.g., 9.77) are significantly worse than those in the official repository (e.g., 2.23). Can you explain this discrepancy? How did you determine the "optimal" guidance scale for the CFG baseline, as the one chosen does not seem to reflect its best performance?

3. Why were different guidance scales used for the Stable Diffusion model in Table 1 versus Table 2? This raises concerns about parameter cherry-picking to favor FDG on specific metrics.

4. The precision/recall scores suggest FDG may be trading fidelity for diversity. Can you provide the Inception Score (IS) results for your experiments to confirm that fidelity is not degraded?

5. How does FDG conceptually and empirically compare to CFG++? Why was this relevant prior work, which also addresses quality at low guidance scales, omitted from your comparisons?

---

> ### Author Response · Authors · 2025-11-14
>
> We thank the reviewer for the time spent reviewing our paper and the provided feedback. We appreciate that the reviewer has pointed out several areas where the clarity of our presentation can be improved, but we note that a number of issues the reviewer raises are already addressed in the current submission. Further, we strongly and respectfully disagree with the suggestion that we may have cherry-picked our results or otherwise performed unfair comparisons, and we hope that the reviewer is positively swayed by our responses to his or her main concerns.
>
> ### **Insufficient Evidence**
> We would like to emphasize that our claims in the paper rest on extensive evidence beyond Figure 2:
>
> - **Quantitative Evaluation**: Figure 5 (systematic ablation isolating the effects of $w_{\text{low}}$ vs. $w_{\text{high}}$), Figure 6 (prompt alignment analysis), Figure 7 (frequency decomposition across steps), Figure 11 (parameter sweeps), and Tables 1-2 (improvements across 6 models).
>
> - **Qualitative results**: Figures 2, 3-4, 9, and 13-15 showing consistent effects across models. Figure 9 (Appendix B.1) explicitly isolates frequency components, showing low frequencies control global structure while high frequencies control details.
>
> If the reviewer has a specific additional experiment in mind, we would be happy to consider adding it.
>
> ### **Lack of Clarity**
> We acknowledge that some experimental details were placed in the appendix due to space constraints, and we are open to reconsidering their placement in the final version of the paper. We always match the CFG scale $w$ with one of the guidance scales in our method ($w_{\text{low}}$ or $w_{\text{high}}$) to ensure a fair comparison across experiments. Appendix D currently provides the full experimental settings. We made a deliberate choice to prioritize comprehensive experimental validation in the main paper (Tables 1–2, Figures 3–8) to demonstrate FDG’s effectiveness across diverse models, datasets, and metrics. That said, we agree that these details are important. The camera-ready version allows up to 10 pages, and we will use this additional space to include further experimental information in the main text.
>
> ### **Unfair Comparison**
> We would like to note that when using CFG in practice, there is *always* a dilemma in choosing the guidance scale: lower scales provide better diversity and color fidelity, while higher scales improve sample quality at the cost of reduced diversity and oversaturation. As a result, there is no single best CFG scale—its effectiveness depends on the specific application and desired outcome. With this in mind, we aimed for a principled evaluation of CFG and FDG across two practical regimes:
>
> - **Table 1 (high CFG):** Standard CFG achieves high precision but poor diversity. FDG preserves precision while improving FID and Recall.
> - **Table 2 (low CFG):** Standard CFG suffers from degraded quality. FDG substantially improves all quality metrics.
>
> Together, these results highlight FDG’s core contribution: it avoids both failure modes of standard CFG across practical guidance scales. Figures 5 and 11 further provide sweeps across a wide range of guidance scales, showing that FDG consistently outperforms CFG and that these conclusions hold across several setups.
>
> We also note that our qualitative comparisons already use CFG parameters that are near-optimal with respect to FID: Figure 3 uses $w=1.25$ (the best FID reported in the EDM2 paper), and Figure 4 uses $w=2.0$ (near-optimal for SDXL according to Figure 12 of the SDXL paper). In both cases, FDG preserves the diversity and colors of CFG while substantially enhancing detail and overall quality.
>
> ### **Question about the reported FID baseline**
> First, we would like to clarify that the objective of our evaluation in Table 1 differs from that of the original papers. While prior work reports results at guidance scales chosen to minimize FID, our goal is to assess performance at commonly used *practical* guidance scales that better reflect real-world usage (e.g., $w=7$ for Stable Diffusion 3). In Table 1, we therefore use higher guidance scales (cfg = 2.0–3.0 for ImageNet models, cfg = 7–10 for text-to-image) that practitioners typically adopt to obtain better sample quality and prompt alignment, even though these scales lead to higher FID.
>
> Additionally, as noted in the appendix, our main evaluations used 10k generated images due to computational constraints. Since all sampling pipelines are compared under identical settings, the comparison remains fair and meaningful. For completeness, we also evaluated CFG and FDG using 50k samples on EDM2-S and obtained the following FID scores (which outperforms the 2.23 reported in the original paper):
>
> |  | FID |
> | --- | --- |
> | CFG ($w=2$) | 4.28 |
> | FDG ($w_{\text{low}}=1$, $w_{\text{high}}=2$) | **1.89** |
>
> We hope this clarifies the rationale behind our evaluation choices, and we will incorporate these explanations into the revised manuscript.

---

> ### Author Response · Authors · 2025-11-14
>
> ### **Potential cherry-picking of parameter**
> We would like to further emphasize that Tables 1 and 2 represent two distinct evaluation setups designed to cover both high and low guidance scales and not an effort to cherry-pick hyperparameters. Figures 5 and 11 additionally provide sweeps across a wide range of scales, showing that FDG consistently outperforms CFG and that these conclusions hold across multiple setups. Taken together, we believe this offers strong evidence that FDG’s superior performance does not depend on any specific guidance scale.
>
>
> ### **Omission of critical metrics**
> We would be happy to include additional metrics in the final version of our paper. Please also note that IS has been shown to have weak correlation with the actual quality of generated images (e.g., see Table 12 of [1]). For this reason, we relied on FID, Precision and Recall to separately measure quality and diversity. Also note that the precision difference between FDG and CFG is minimal (0.85 vs. 0.83), indicating that FDG improves diversity without trading off quality. We would be happy to clarify this further if the reviewer has additional questions.
>
> [1] Sadat S, Buhmann J, Bradley D, Hilliges O, Weber RM. CADS: Unleashing the diversity of diffusion models through condition-annealed sampling. arXiv preprint arXiv:2310.17347. 2023 Oct 26.
>
> ### **Comparison with CFG++**
> CFG++ focuses on correcting the off-manifold drift that arises at high guidance scales (especially for low-NFE models and image-editing tasks), and it does not evaluate the diversity and saturation issues of CFG. In contrast, our work analyzes CFG behavior in the frequency domain, highlighting how different frequency bands influence the final generations. Also note that CFG++ has a different definition for the guidance scale: as mentioned in that paper, a scale of 0.6 for CFG++ corresponds to a CFG scale of 7.5 for Stable Diffusion, which lies in the high-scale regime we study in Table 1. We would be happy to include this discussion in the final version of our paper.

---

> ### Comment · Reviewer_qzZU · 2025-11-25
>
> Thank you for your clarification. I have further questions.
>
> **Regarding CFG Scales in Tables 1 and 2**
>
> It seems that in Table 1, you report the results for the high CFG scale, while in Table 2, the results are reported for the low CFG scale. However, if the intention is to fairly compare CFG and FDG across both low and high CFG scales, then both experiments should have been conducted with both settings. According to your approach, this would mean that the results in Table 1 should also be provided for the low CFG scale, and the results in Table 2 should be provided for the high CFG scale. Could you provide the results in Table 1 for the low CFG scale and in Table 2 for the high CFG scale to ensure consistency in your comparisons?
>
> **Regarding Inception Score (IS)**
>
> While you mention that IS has weak correlation with the actual quality of generated images, it remains a widely used metric in the field. Given its prevalence and its role in evaluating the fidelity of generated samples, could you provide the Inception Score (IS) results for your experiments? Reporting IS would allow for a more complete understanding of the fidelity-diversity trade-off, which is crucial for evaluating the true performance of the models in your study.

---

> > ### Author Response · Authors · 2025-11-26
> >
> > We thank the reviewer for their engagement. Please find our responses to the questions below, and we remain open to further discussion.
> >
> >
> > ### **Comparisons in Table 1 and 2**
> > Figure 5 in our submission already reports the metrics listed in Table 1 across guidance scales (FID, precision, recall, and saturation). We have additionally extended this comparison to human preference scores using Stable Diffusion 3 to further highlight the advantages of FDG over CFG at different guidance strengths. The full results are provided [here](https://ibb.co/VcRwYrMW) and [here](https://ibb.co/8DdN9H9k). Across all settings, FDG delivers better quality metrics (e.g., HPSv2) while avoiding common CFG failure modes such as reduced diversity and oversaturation.
> >
> > ### **Inception Score**
> > The corresponding Inception Score plots for CFG vs. FDG are available [here](https://ibb.co/YFCMhQpW). IS increases with guidance scale for both methods; however, IS is known to be unreliable when evaluating diverse outputs. Because CFG reduces diversity, it artificially inflates IS. This effect is evident when computing IS on the ImageNet Eval subset (dashed line): although the images come from real data, the score is substantially lower than that of CFG.

---

### Official Review · Reviewer_Db2d · 2025-10-30

**Soundness:** 3
**Presentation:** 3
**Contribution:** 2
**Rating:** 4
**Confidence:** 3

**Summary:**

This paper addresses a well-known and critical trade-off in conditional diffusion models that utilizes the classifier-free guidance (CFG). The authors present a novel analysis of this problem by decomposing the CFG update signal in the frequency domain. The key insight is that low-frequency (LF) and high-frequency (HF) components of the guidance signal have distinct and opposing effects. The authors find that LF-CFG governs global structure and condition alignment. However, LF-CFG is the main cause of diversity loss and color oversaturation. On the other hand, HF-CFG primarily enhances visual fidelity and fine details. Based on this, the authors propose frequency-decoupled guidance (FDG) that decomposes the CFG signal into LF and HF parts and amplifies the HF-CFG signal to generate high-fidelity samples while maintaining the diversity. Experiments on diverse diffusion models prove the effectiveness and generalizability of the FDG.

**Strengths:**

- **Simplicity**: FDG is a plug-and-play and training-free method that only introduces a few additional lines of code with negligible computational cost. This makes FDG a versatile method for general conditional diffusion models.

- **Comprehensive Evaluation**: The authors evaluate FDG's compatibility with various base diffusion models (EDM, SD, etc.), various samplers (DDIM, DPM++, etc.), distilled models (SDXL-Lightning), and other guidance-improvement techniques (CADS, APG, FreeU), showing the robustness of the method.

- **Principled Analysis on Guidance**: The authors analyze CFG from the perspective of the frequency domain. In addition, the authors provide a theoretical analysis of why auto guidance (AG) has shown superior performance compared to CFG. This paves the way to analyze the effectiveness of the guidance, which previously heavily relied on the output sample quality.

**Weaknesses:**

- **Discrepancy in FID Reporting**: There appears to be a mismatch between the reported baseline CFG FIDs for models in Table 1 and the FIDs commonly reported in their respective original papers, like EDM2-S (9.77 vs. 2.23), EDM2-XXL (8.65 vs. 1.81), and DiT-XL/2 (9.31 vs. 2.27). While I'm not comparing all the other models, it seems inconsistency of FIDs may stem from the arbitrary guidance scale (e.g., DiT-XL utilizes cfg=1.5 for the best FID, but the paper reports FID with cfg=2.0 according to Table 12) and incomprehensive evaluation metric (i.e., the paper did not report IS which is common metric to measure the sample quality in c2i image synthesis).

- **Details of Evaluation**: In addition to the above weakness, the evaluation setup is not principled. Specifically, for Table 1, baseline CFG is set to $w_{high}$ of FDG, while for Table 2, baseline CFG is set to $w_{low}$. I suspect that this inconsistency may stem from the fact that Table 1 usually reports the metrics regarding the diversity, where low CFG is beneficial, while Table 2 usually reports the metrics regarding the sample quality, where high CFG is beneficial. This suggests the baselines are not being evaluated fairly or in their optimal settings. This inconsistent methodology severely damages the faithfulness of the paper's comparisons.

- **Limited Applicability to Diffusion Models**: The Recent visual generative modeling community actively researches autoregressive models (e.g., LlamaGen, VAR) and masked generative models (MaskGIT, MaskGIL) for compatibility and seamless integration with LLMs.
which often utilize discrete visual tokens and perform CFG in a logit space.

**Questions:**

- It seems in Figure 2, FDG also shows color saturation in the dog in the bottom right.
- The method's robustness to the frequency decomposition process itself is not adequately addressed. FDG relies on separating HF and LF signals, a process that is highly sensitive to the specifics of the low-pass filter (e.g., the Gaussian kernel's size and $\sigma$). The paper lacks an ablation study on how different kernel parameters would affect FDG's performance.
- $w_{low}$ and $w_{high}$ are key hyperparameters to balance the frequency components. Is there a principled approach or a systematic heuristic for tuning these weights to best navigate the sample quality vs. diversity trade-off?

**Details Of Ethics Concerns:**

No ethical concerns raised.

---

> ### Author Response · Authors · 2025-11-14
>
> We thank the reviewer for providing feedback on our submission, as well as for recognizing the contributions of our work. Please find our detailed responses below, and we would be glad to engage in further discussion if any questions or concerns remain.
>
> ### **Discrepancy in FID Reporting**
> First, please note that our experimental goal in Table 1 differs from the original papers. The original papers report results at guidance scales that minimize FID, whereas we evaluate each model at commonly-used practical guidance scales to demonstrate FDG's effectiveness in real-world scenarios (e.g., $w=7$ for Stable Diffusion 3).
>
> We use higher guidance scales in Table 1 that practitioners typically employ for better quality and prompt alignment, despite higher FID ($w=2.0,3.0$ for ImageNet models, and $w=7,10$ for text-to-image generation). This is precisely the problem FDG addresses: at these practical guidance scales, standard CFG suffers from poor FID due to reduced diversity. FDG achieves substantially better FID at these same practical scales, showcasing high-quality generations with significantly better diversity. We also provide sweeps across various guidance scales in Figures 5 and 11, demonstrating that even when comparing the best FID of each method, FDG consistently outperforms CFG.
>
> Finally, as stated in the appendix, we conducted our evaluations using 10k generated images due to computational constraints. Since our experiment compares different sampling pipelines under identical experimental conditions, the comparison remains fair and meaningful. When evaluating CFG and FDG using 50k samples with EDM2-S, we obtain the following FID scores (which is also better than 2.23 reported by the paper):
>
> |  | FID |
> | --- | --- |
> | CFG ($w=2$) | 4.28 |
> | FDG ($w_{\text{low}}=1$, $w_{\text{high}}=2$) | **1.89** |
>
> We hope this answers the reviewer’s concern, and we will clarify these points in our current draft.
>
> ### **Details of Evaluation**
> We are happy to further clarify the reasoning behind the experiments in Tables 1 and 2, and why we believe they represent a fair and principled evaluation. We respectfully claim that there is no inconsistency here. Our design intentionally evaluates two practical scenarios that highlight how FDG resolves the fundamental trade-off in CFG:
>
> - **Table 1 (high CFG):** Standard CFG achieves high precision but poor diversity. FDG preserves precision while improving FID and Recall.
> - **Table 2 (low CFG):** Standard CFG suffers from degraded quality. FDG substantially improves all quality metrics.
>
> Together, these results illustrate FDG’s core contribution: it avoids both failure modes of standard CFG across practical guidance scales. Figures 5 and 11 provide full sweeps across various guidance scales, showing that even at CFG’s optimal settings, FDG consistently achieves better FID.
>
> ### **Limited applicability to diffusion models**
> FDG targets continuous diffusion models, which remain the dominant paradigm for state-of-the-art image generation (such as Stable Diffusion). While autoregressive and masked models are emerging, they represent a different modeling paradigm with discrete tokens. Extending frequency-based guidance to the logit space is an interesting direction for future work.
>
> ### **Robustness to the frequency decomposition process**
> We provide ablation studies on the frequency decomposition operator in Appendix C, Table 11. Table 11(a) compares Laplacian pyramids and wavelet transforms, showing both yield similar performance (FID of 5.12 vs 5.26). Table 11(b) explores multi-band frequency decomposition, again yielding very similar results. These ablation experiments demonstrate that FDG is not sensitive to the specific decomposition method as long as it provides meaningful frequency separation.
>
> Regarding kernel parameters (e.g., Gaussian kernel size): we use the standard Laplacian pyramid implementation with default parameters from established libraries (Kornia).
>
> ### **Systematic heuristic for tuning the weights in FDG**
> In Figure 11, we perform an ablation study exploring different values of $w_{\text{high}}$ and $w_{\text{low}} = r(w_{\text{high}} - 1) + 1$ with $w_{\text{high}} \in \\{1.5, 2, 2.5\\}$. The results show that setting $r < 0.5$ works well across different $w_{\text{high}}$ values, consistently improving FID and recall while maintaining precision. This provides a principled heuristic: choose any reasonable $w_{\text{high}}$ (e.g., 1.5-3 for ImageNet, 7-15 for text-to-image generation) and set $w_{\text{low}} ≤ 0.5(w_{\text{high}} - 1) + 1$.
>
> In practice, like standard CFG tuning, we select values by generating a few samples and visually inspecting them. Importantly, Figures 5 shows FDG outperforms CFG across most of the parameter space, demonstrating robustness to hyperparameter choices.

---

> > ### Comment · Reviewer_Db2d · 2025-11-27
> >
> > After reading the author's response, I keep my original score for the following reasons:
> >
> > - The CFG is widely adopted for the visual generative models, while FDG is limited to continuous diffusion models.
> > - While the authors provide some reasons for the experimental setup for Tables 1 and 2, it is still not comprehensive. As noted in the first review, Table 1 usually reports the metrics regarding the diversity, where low CFG is beneficial, while Table 2 usually reports the metrics regarding the sample quality, where high CFG is beneficial. The authors should include (1) both CFG results that correspond to the $\omega_{high}$ and $\omega_{low}$ of FDG, (2) IS for Table 1, and (3) FID for Table 2.
> > - As illustrated in Figure 5, FDG achieves better FID and Recall, while showing lower Precision. From these results, I suspect that with comprehensive experiments, FDG will not show clear improvement over naive CFG with scaling adjustment.

---

> ### Author Response · Authors · 2025-11-27
>
> We thank the reviewer for the engagement in the rebuttal and for giving us the opportunity to clarify these final points. We address the remaining concerns below, and we hope the clarifications and additional evidence help the reviewer confidently revise their assessment upward.
>
> ### **Applicability to continuous diffusion models**
> While we acknowledge the growing interest in autoregressive and discrete models, continuous diffusion models (such as Stable Diffusion, and EDM) remain the dominant paradigm for state-of-the-art image generation. FDG is designed to solve a fundamental issue (oversaturation and diversity loss) within this widespread framework. If the reviewer has a specific discrete model in mind, we would be happy to include a discussion of that in the final version.
>
> ### **Comprehensiveness of Tables 1 & 2**
> The reviewer requested "low-scale" results for the Table 1 setup and "high-scale" results for the Table 2 setup. We would like to clarify that Figure 5 in our submission already extends the metrics in Table 1 across the full spectrum of guidance scales. Furthermore, to fully address the reviewer's request regarding Table 2, we have extended our experiments to plot performance across the full range of guidance scales (low to high) for the human preference scores. The complete results can be viewed here:
>
> - **Table 1 Sweep:** https://ibb.co/VcRwYrMW
> - **Table 2 Sweep:** https://ibb.co/8DdN9H9k
>
> **Key Takeaway:** Across all settings, FDG consistently delivers superior quality metrics (e.g., FID and HPSv2) while avoiding the well-known failure modes of standard CFG, such as reduced diversity and oversaturation.
>
> ### **The request for FID in Table 2**
> We cannot report FID in Table 2 because the datasets and prompts used in Table 2 do not have the standard reference statistics required for a valid FID calculation.
>
>
> ### **Inception Score (IS) Analysis**
>
> Per your request, we have generated the Inception Score plots comparing CFG and FDG, available [here](https://ibb.co/YFCMhQpW). However, please note that while IS increases with the guidance scale for both methods, we strongly caution that IS is notoriously unreliable when evaluating diverse outputs. Because CFG reduces diversity, it artificially inflates the Inception Score. This is empirically demonstrated in our plot: the dashed line represents the ImageNet Eval subset (real data). Despite being "perfect" real images, their IS is substantially lower than that of high-scale CFG. This confirms that a higher IS in this context often signals a loss of diversity rather than an increase in quality.
>
> ### **FDG vs. "Naive" CFG Tuning**
> The reviewer expressed a suspicion that FDG might not offer a clear improvement over simply tuning the scale of CFG. We respectfully but firmly disagree, based on the empirical evidence in Figure 5, Figure 11, as well as the new plots linked above. The results provide strong evidence that FDG offers the following advantages:
> - **Shifting the Pareto Frontier:** The parameter sweeps show that FDG does not merely move along the same curve as CFG. Instead, the _best_ achievable quality metrics (such as FID and HPSv2 scores) with FDG are consistently better than the _best_ achievable scores under CFG at any scale. Moreover, FDG outperforms CFG across a broad range of guidance scales in both plots.
>
> - **Marginal Cost vs. Significant Gain:** The drop in precision that the reviewer mentioned is negligible (e.g., 0.85 vs. 0.83), whereas the gains in FID, Recall, and Human Preference Scores are substantial.
>
> **Conclusion:** This confirms that FDG pushes the performance of guidance beyond the limits of naive CFG tuning, rather than simply trading one metric for another.

---

### Official Review · Reviewer_QYwU · 2025-11-01

**Soundness:** 3
**Presentation:** 2
**Contribution:** 3
**Rating:** 4
**Confidence:** 3

**Summary:**

This paper empirically observes that applying strong classifier-free guidance (CFG) to low-frequency components often leads to over-saturation and reduced sample quality. To address this, the authors propose decomposing each predicted image into low- and high-frequency components at every inference step and applying a stronger CFG scale to the high-frequency components in the frequency domain. This approach mitigates the inherent trade-off in CFG between diversity and quality, where low CFG scales yield high diversity but low quality, and high CFG scales produce low diversity but high quality, while also alleviating the saturation problem observed at high guidance scales. The method achieves this improvement without increasing the number of model predictions required per sampling step, using only a simple Laplacian transform for frequency separation.

**Strengths:**

- This paper tackles one of the key issues in CFG, saturation at high guidance scales, and provides a principled way to mitigate it.
- Unlike other guidance methods that require multiple model predictions (often 2×2 = 4 predictions) to combine different forms of guidance, the proposed approach enhances sample quality without additional model calls, maintaining computational efficiency.
- The use of a lightweight frequency-domain decomposition (via Laplacian transform) makes the approach conceptually clean and easy to integrate with existing diffusion frameworks.

**Weaknesses:**

- Lack of proper citation. In the Introduction (L084-L098), the discussion lacks references, even though prior works have already analyzed diffusion processes in the frequency domain, such as FreeU (Si et al., 2024) and [a]. These should be cited appropriately to situate this work in the existing literature.
- Ambiguity in frequency computation. It is unclear how the frequency components are calculated. Since intermediate predictions are inherently noisy, it should be explicitly stated whether the decomposition is applied directly to the noisy intermediate prediction or to the denoised $\hat{x}_0$. This detail appears only in the pseudocode or Supplementary Material D, which makes it difficult for readers to follow. It would improve clarity to include this explanation in the main method section.

    Moreover, as pointed out by [a], high-frequency components tend to be destroyed in high-noise regions during the diffusion process. The paper should therefore discuss how meaningful frequency separation is achieved when the predicted $\hat{x}_0$. may already lose high-frequency information in these regions.

- Missing comparison with self-attention guidance. The paper should discuss and cite Self-Attention Guidance [b]. In [b], the authors blur high-frequency regions in the final image to create degraded samples for guidance, improving quality, conceptually, the opposite of enhancing high-frequency guidance as proposed here. Interpreting [b] from the frequency-domain perspective of this work could yield valuable insights, and such a discussion would enrich the paper.
- Experimental details are overly deferred to the Supplementary Material. Important information, such as experimental settings, resolution for ImageNet evaluation, and key hyperparameters (e.g., CFG scale used for baselines, FID10K vs FID50K), should appear in the main paper. While visual results are helpful, some could be moved to the supplement to make room for these critical quantitative details.

[a] Dieleman, Sander, Diffusion is spectral autoregression, 2024

[b] Hong, Susung, et al., Improving sample quality of diffusion models using self-attention guidance. ICCV 2023

**Questions:**

I believe the overall presentation could be significantly improved. The number of qualitative result figures could be reduced, and the details discussed in the weaknesses section should be more thoroughly incorporated into the main text to make the paper more solid and convincing.

---

> ### Author Response · Authors · 2025-11-14
>
> We appreciate the reviewer’s feedback and the recognition of the contributions made in our work. Below, we address each of the comments in detail and welcome any further discussion or clarification.
>
> ### **Lack of proper citation**
> Please note that we discussed FreeU in Appendix B.9 and also showed that its effects are complementary to FDG. FreeU operates by modifying UNet architecture features, whereas FDG focuses on analyzing the frequency components of the CFG guidance signal. In addition, [a] connects diffusion and autoregressive models through frequency analysis, while our work focuses specifically on improving CFG through frequency decoupling. We would be happy to clarify these distinctions in the revised introduction.
>
> ### **Ambiguity in frequency computation**
> As stated in L147, our diffusion formulation in Section 3 is based on the denoised prediction $\hat{x}_0$ (i.e., $D(z_t, t)$ approximates the clean signal $x$ from the noisy signal $z_t$). We also followed this notation in our implementation and used the denoised predictions to guide generation. We will make this clearer in the revised manuscript as well.
>
> ### **Low-frequency components in high-noise regions**
> The dominance of low-frequency components in CFG at early sampling is precisely why FDG performs better than CFG. We would like to elaborate on this point. Since the frequency decomposition is applied to the predicted image $\hat{x}_0$, this prediction naturally contains mostly low-frequency content in the early steps when the noise level is high. Figure 7 further confirms that in high-noise regions, the CFG signal is dominated by low-frequency components, while the high-frequency components have smaller norms. Our analysis shows that this imbalance is detrimental, as strong guidance on lower frequencies pushes samples toward less diverse and more saturated outputs. FDG enhances this behavior by reducing low-frequency guidance and amplifying high-frequency signals, even when the latter have smaller magnitudes.
>
> ### **Comparison with self-attention guidance**
> We thank the reviewer for this suggestion. In Appendix B.7, we compared FDG with perturbed-attention guidance (PAG) [1], which is an improved and more recent version of Self-Attention Guidance. Table 7 demonstrates that FDG and PAG are complementary, and combining them leads to further improvements over PAG alone.
>
> ### **Question about experimental details**
> We acknowledge that some experimental details are in the appendix due to space constraints. We made a deliberate choice to prioritize comprehensive evaluation in the main paper (Tables 1-2, Figures 3-8) to demonstrate FDG's effectiveness across diverse models, datasets, and metrics. That said, we agree these details are important. The camera-ready version allows up to 10 pages, and we will use this additional space to include further experimental details in the main text.
>
> [1] Ahn D, Cho H, Min J, Jang W, Kim J, Kim S, Park HH, Jin KH, Kim S. Self-rectifying diffusion sampling with perturbed-attention guidance. InEuropean Conference on Computer Vision 2024 Sep 29 (pp. 1-17). Cham: Springer Nature Switzerland.

---

### Official Review · Reviewer_3MXD · 2025-11-01

**Soundness:** 2
**Presentation:** 3
**Contribution:** 3
**Rating:** 4
**Confidence:** 4

**Summary:**

The authors analyze classifier-free guidance in diffusion models from the perspective of frequency domain. They argue that by decoupling guidance for higher-freqencies from lower-frequencies by using different weights in each, they can achieve the best performance. An extensive set of experiments is provided.

**Strengths:**

1. I like the presentation of the paper. The motivation, figures, and explanations are clear.
2. The empirical results are suggesting that the proposed method significantly improves the standard classifier-free guidance
3. I appreciate the authors take on explaining the performance of autoguidance and guidance in limited time intervals. Seems convincing to me.

**Weaknesses:**

The biggest weakness of the paper, in my opinion, is around the question of whether the comparison with CFG is fair. The authors say (line 310): "hyperparameters used for each experiment are given in Appendix D", and they are provided in Table 12. However, the procedure for selecting these hyperparameters: "the guidance scales are selected in the same way practitioners typically choose CFG values for a model, i.e., generating a few samples and visually inspecting them" makes me doubt that the comparison is fair. The reported improvements for multiple (almost all) metrics in tables 1, and 2 are extremely large, and much larger than what I would expect based on Figure 5. The paper does not convince me that the hyperparameters were properly tuned for CFG to produce the results in Tables 1, and 2. I have no reason to believe that the performance gaps would be much smaller if I tuned the parameters for CFG. An experiment that would convince me would be: perform a sweep over CFG weights, and compare the best performing ones with FDG.

I think that the proposed idea is very promising, and I am prepared to raise my score if the authors present convincing evidence that the reported improvements are not an artifact of poorly tuned CFG weights.

**Questions:**

1. Can the authors perform a quantitative comparison (as in Tables 1, 2) but find the best parameters for CFG? Then, in qualitative comparisons (e.g. Figures 3 and 4), use the CFG parameter that actually performed best?
2. It looks like FDG hurts precision, which aims to measure "how realistic" the samples are. Figure 5 shows that the proposed approach is consistently worse than the standard CFG. Similarly, figure 11. Can the authors comment on this? It feels like an important metric that shouldn't be disregarded.
3. I don't think I understand Figure 6. To me, it looks like the blue curve (standard CFG) achieves the best performance, and the green curve corresponds to the proposed approach (stronger guidance for high frequencies than low). What choice of parameters does the dotted line correspond to? Either way, the figure clearly suggests that the same performance can be obtained with the standard CFG.
4. The authors say that the method "does not incur any noticeable computational overhead". Can this statement be made more precisely? Is the overhead 10% or 0.01%? Just to be clear, I am not asking for any additional experiments here, just curious.
5. Usually, increasing the CFG weight makes FID worse, but improves the inception score (IS). People usually evaluate multiple CFG weights and look at the "pareto curves", i.e. FID on one axis and IS on the other. Can the authors produce such a curve for regular CFG, and show where on the FID/IS graph the proposed approach lands?

---

> ### Author Response · Authors · 2025-11-14
>
> We thank the reviewer for the evaluation of our paper, and for highlighting its key strengths. Below, we provide detailed responses to each of the comments, and we would be glad to further clarify any remaining questions or concerns.
>
> ### **Fair comparison with CFG**
> We thank the reviewer for raising this question and would like to clarify it further. First, please note that we always match the CFG scale $w$ with one of the guidance scales in our method ($w_{\text{low}}$ or $w_{\text{high}}$) to ensure a fair comparison across experiments. To make our setup clear, we emphasize that when CFG is used in practice, there is always a dilemma in choosing the guidance scale: lower scales offer better diversity and color fidelity, while higher scales improve sample quality at the cost of reduced diversity and oversaturation. Our experiments are designed to show that FDG avoids these issues.
>
> Accordingly, we compare the performance of CFG and FDG under two regimes: high CFG scales (e.g., 7 for SDXL, the default) and low CFG scales (e.g., 3 for SDXL). Table 1 reports results for the high-scale setup, and Table 2 for the low-scale setup. We show that for higher CFG scales typically used in practice, FDG achieves similar precision (e.g., 0.85 vs. 0.83) while significantly improving FID and Recall. Conversely, for lower CFG scales, FDG substantially improves all quality metrics for Stable Diffusion models, as evidenced in Figure 4 and Table 2. Figures 5 and 11 also include additional experiments sweeping the guidance scales in CFG and FDG, showing that the best FID achieved by FDG consistently outperforms the best CFG result. Taken together, these observations provide strong evidence that FDG is a superior alternative to CFG across a wide range of guidance scales, improving quality over unguided generation while avoiding the common pitfalls of CFG.
>
> ### **Comparison with best CFG parameters (Fig 3 and 4)**
> Our main qualitative comparisons already used optimal CFG parameters w.r.t. FID: Figure 3 uses $w=1.25$ (the best FID value reported in the EDM2 paper), and Figure 4 uses $w=2.0$ (near-optimal for SDXL according to Figure 12 of the SDXL paper). In both cases, FDG preserves the diversity and colors of CFG while substantially enhancing detail and overall quality. We will clarify the design choice of the parameters in the revised manuscript.
>
> ### **Effect of FDG on Precision**
> Please note that the plot in Figure 5 is scaled for better visuals, and the actual difference between the precision values are minor (0.85 vs 0.83). We also report other quality metrics such as ImageReward and HPSv2 in Table 2, which we believe are better aligned with human judgment of individual image quality.
>
> ### **Clarification for Figure 6**
> Figure 6 illustrates the impact of the high- and low-frequency components in CFG on prompt alignment. We show that both components contribute to higher CLIP scores, with the low-frequency component having the stronger effect. The dotted line corresponds to the FDG setup ($1 < w_{\text{low}} < w_{\text{high}}$), while the green and orange lines represent $w_{\text{low}} = 1$ and $w_{\text{high}} = 1$, respectively, to isolate the role of high and low frequencies. This also shows that the FDG setup maintains the prompt alignment of very high CFG scales while avoiding their common drawbacks, such as oversaturation and reduced diversity.
>
> ### **Computational cost of FDG**
> As discussed in Appendix C, we tested the inference throughput of FDG and CFG using Stable Diffusion 3, and both methods achieved the same sampling speed (1.22 iterations per second). We thus conclude that FDG has negligible impact on inference time, as nearly all sampling computations are dominated by forward passes through the model.
>
> ### **Pareto curves**
> We thank the reviewer for bringing up this question, and we will include the corresponding Pareto front curves in the final version of the paper. Please also note that this is implicitly reflected in Figure 5, where flatter FID curves for FDG indicate better performance over a wider range of guidance scales (which in turn results in a broader usable range in practice).

---

> > ### Comment · Reviewer_3MXD · 2025-11-27
> >
> > I thank the authors for their response. I understand that CFG weight is usually heuristically chosen. What I meant was:
> >
> > Can the authors demonstrate that their proposed choice of high- and low-frequency cfg weights outperforms vanilla CFG for a range of CFG values?
> >
> > I am inclined to accept the paper, because I am impressed with the empirical results, and I particularly appreciate the simplicity of the method, so I will raise my score.

---

### Author Response · Authors · 2025-12-03
**Summary of the rebuttal**

We thank all reviewers for their constructive feedback and thoughtful evaluation of our work. Below, we summarize the main concerns raised and our responses.

### **Summary of reviews**

All reviewers acknowledged the novelty of our frequency-domain analysis of CFG and the practical value of FDG as a plug-and-play, training-free method. The main concerns were: (1) fairness of baseline comparisons and hyperparameter selection, (2) discrepancies in reported FID values compared to original papers, and (3) missing metrics (Inception Score). We provided detailed responses in the rebuttal, and we summarize them below.


### **Fair comparison with CFG**
We emphasize that Figures 5 and 11 in our submission provide **comprehensive parameter sweeps across the full range of guidance scales**, showing that FDG consistently outperforms CFG throughout the parameter space—not only at isolated, cherry-picked values. To further address reviewer concerns, we performed additional parameter sweeps for human preference scores. The full results are available [here](https://ibb.co/VcRwYrMW) and [here](https://ibb.co/8DdN9H9k). Across all metrics (FID, recall, saturation, ImageReward, HPSv2, and CLIP Score), FDG consistently outperforms the best performance of CFG at any guidance scale while maintaining nearly identical precision.

Additionally, CFG has no single optimal scale: lower scales offer better diversity but degraded quality, while higher scales improve quality but reduce diversity and cause oversaturation. Tables 1 and 2 in our submission intentionally evaluate these two regimes:
- **Table 1 (high CFG):** Common practical scales ($w=7$ for SDXL, $w=2-3$ for ImageNet) where CFG achieves high precision but poor diversity
- **Table 2 (low CFG):** Lower scales where CFG suffers from degraded quality
This demonstrates that FDG avoids both failure modes across the practical range of guidance scales.

### **Differences in reported FID**
The original DiT and EDM2 papers report FID at CFG scales optimized specifically for FID. However, Table 1 in our paper explicitly evaluates the models at practical CFG scales (e.g., $w=7$ for SD3) reflecting real-world usage. We also used 10K samples for computational efficiency, as noted in Appendix D. When using 50K samples on EDM2-S, we obtain the following results (which also improve upon the FID of 2.23 reported in the EDM2 paper):

|  | FID |
| --- | --- |
| CFG ($w=2$) | 4.28 |
| FDG ($w_{\text{low}}=1$, $w_{\text{high}}=2$) | **1.89** |

### **Additional metrics and comparisons**
* **Inception Score:** We provided the IS plot [here](https://ibb.co/YFCMhQpW). However, please note that IS is unreliable when evaluating diverse outputs. CFG artificially inflates IS by reducing diversity, causing the evaluation subset of ImageNet (dashed line) to score lower than high-scale CFG.

* **Precision values:** We clarified that the precision drop mentioned by reviewers is negligible (0.85 vs. 0.83), whereas FDG yields substantial improvements in FID, recall, and human preference scores. This strongly indicates that FDG avoids CFG’s typical diversity failures without compromising precision.

* **Extended comparison for human preference scores:** We added full sweeps for human preference scores (links above), further confirming FDG’s consistent superiority over CFG across guidance scales.

### **Final Comment**
We believe our comprehensive evaluation, including parameter sweeps across all guidance scales (Figures 5, Figure 11, https://ibb.co/VcRwYrMW, and https://ibb.co/8DdN9H9k), experiments on 6 diverse models (Tables 1-2), and extensive qualitative results provides strong evidence that FDG offers genuine improvements over CFG beyond simple hyperparameter tuning. FDG shifts the Pareto frontier, achieving better FID, recall, and human preference scores than the best possible CFG at any scale, while maintaining almost the same precision. Importantly, FDG avoids CFG's inherent limitations (reduced diversity and oversaturation at high scales, degraded quality at low scales) and outperforms CFG consistently across a wide range of guidance values.

---

### Meta-Review · Area_Chair_V3G9 · 2026-01-05

**Summary:**

This paper proposes a method to improve image generation by applying different guidance strengths to low and high frequencies. Reviewers appreciated the simple idea and the strong results across various models. However, they raised serious concerns about unfair comparisons and missing experimental details. Reviewers specifically noted that the baseline scores for the standard method seemed incorrect or poorly tuned. The authors provided more data during the discussion, but the disagreement regarding fair baselines persisted after the rebuttal. Overall, the paper is less convincing, and the authors are encouraged to fix the baseline comparisons for future submissions.

**Reviewer Concerns:**

The rebuttal successfully clarified how the method works and addressed missing citations. However, the major concern regarding unfair baseline comparisons remains outstanding. The reviewers still believe the authors need to show consistent results across all settings and include missing metrics like the Inception Score.

**Reviewer Scores:**

Reviewer 3MXD would likely raise their score as they were impressed by the simple solution and additional data. Reviewer QYwU might raise their score since their questions on citations and math were answered. Reviewer Db2d would likely keep their score low because they felt the experiments were still not comprehensive. Reviewer qzZU would maintain a rejection score as they strongly believed the comparisons were unfair and missing key data.

---

### Decision · Program_Chairs · 2026-01-26

Reject